# Effectiveness and Components of Health Behavior Interventions on Increasing Physical Activity Among Healthy Young and Middle-Aged Adults: A Systematic Review with Meta-Analyses

**DOI:** 10.3390/bs14121224

**Published:** 2024-12-19

**Authors:** Jiawei Wan, Jihoon Kim, Takehiko Tsujimoto, Ryoko Mizushima, Yutong Shi, Kosuke Kiyohara, Yoshio Nakata

**Affiliations:** 1Graduate School of Comprehensive Human Sciences, University of Tsukuba, 1-1-1 Tennodai, Tsukuba 305-8574, Japan; s2230479@u.tsukuba.ac.jp (J.W.); s2230477@u.tsukuba.ac.jp (Y.S.); 2Institute of Health and Sport Sciences, University of Tsukuba, 1-1-1 Tennodai, Tsukuba 305-8574, Japan; jihoon.kim.ft@u.tsukuba.ac.jp (J.K.); mizushima.ryoko.gu@u.tsukuba.ac.jp (R.M.); 3Faculty of Human Sciences, Shimane University, 1060 Nishikawatsucho, Matsue 690-8504, Japan; tsujimoto@hmn.shimane-u.ac.jp; 4Department of Food Science, Faculty of Home Economics, Otsuma Women’s University, 12 Sanbancho, Chiyoda, Tokyo 102-8357, Japan; kiyohara@otsuma.ac.jp

**Keywords:** physical activity, health intervention, systematic review, meta-analysis, meta-regression, behavior change

## Abstract

Physical inactivity contributes to various health risks; however, approximately one-third of the global population remains insufficiently active. Many researchers have attempted to increase physical activity levels; however, the effectiveness and the specific components of these interventions remain unclear. This systematic review with meta-analyses utilized a behavior change technique taxonomy to identify and extract effective intervention components, aiming to develop more efficient programs to promote physical activity. We searched the PubMed and Ichu-shi Web databases for studies targeting healthy young and middle-aged adults with data on physical activity changes and extracted the intervention components. A random-effects model was used for the primary meta-analysis, and a meta-regression was conducted for the selected outcomes. Overall, 116 studies were included, with 102 used for the primary analysis. The interventions showed a small effect on overall physical activity promotion. Subsequent meta-regressions identified 1.5 Review behavior goal(s) as a significant positive intervention component, as well as four and three potential positive and negative components, respectively. This systematic review and meta-analysis demonstrated the effectiveness of physical activity interventions and highlighted effective and negative components. These findings may inform the design of future programs aimed at promoting physical activity.

## 1. Introduction

Physical inactivity is a major public health concern, leading to all-cause mortality, cardiovascular diseases, diabetes, and obesity [1]. Longitudinal studies have shown that increasing physical activity (PA) levels can reduce the 5-year mortality risk in adults with obesity [2], the number of cardiovascular disease risk factors [3], and the prevalence of various cancers [4]. To address this concern, the World Health Organization recently recommended that adults engage in at least 150–300 min of moderate aerobic PA per week (or the equivalent amount of vigorous activity) [1]. The working-age population comprises approximately 65% of the total population [5], playing a crucial role in societal productivity and development. Within this demographic, young and middle-aged adults form the majority, making efforts to increase their PA to improve health and prevent diseases highly critical for society. A study utilizing data from the National Health and Nutrition Examination Survey reported that 47.0% and 59.8% of individuals aged 18–44 and 45–64 years, respectively, did not meet the recommended PA levels in the United States [6]. Although some young and middle-aged adults appear healthy, physical inactivity exposes them to high health risks. Thus, increasing PA before health problems emerge is crucial for ensuring sustained health benefits in the early stage and has long been widely recognized [7]. Numerous studies exist on promoting PA among young and middle-aged adults [8,9,10,11,12,13]; however, the evaluation of their overall effectiveness remains limited.

Over recent decades, numerous health behavior interventions have been implemented, and the effectiveness of increasing PA has been examined [14,15]. However, the effectiveness of these interventions varies widely based on their components, delivery methods, and target populations. Identifying effective components is crucial for developing strategies that can successfully drive behavior change. The behavior change technique (BCT) taxonomy for intervention component classification, developed by approximately 54 experts from seven countries and across multiple fields, is suitable for defining the specific components and determining the most effective components in behavior change interventions [16,17]. The BCT is defined as the observable, replicable, and irreducible component of the intervention designed to alter or redirect causal processes that regulate behavior [17]. The development of the BCT taxonomy was based on published technical theories and utilized a structured decision-support technique known as the Delphi method. Through iterative rounds of opinion collection, integration, reliability testing, and feedback, a systematic taxonomy with a high consensus and reliability was achieved [17]. The theoretical application of the BCT has been confirmed to effectively promote improvements in various health behaviors, including PA [18,19,20].

Currently, there is a paucity of research that comprehensively examines the impact of intervention measures on PA levels among healthy young and middle-aged adults. While some studies investigated the effectiveness of BCTs on enhancing PA, their scope is frequently restricted to populations [16,21,22,23], intervention formats [20,24,25], or specific methods [26,27]. Among these studies, only one distinguished between the types of PA [24], whereas the remaining studies did consider these differences [20,21,22,25,26,27]. A multivariate meta-regression can effectively quantify the impact of each BCT on PA; however, only a few studies have employed this approach. Due to the limited number of included studies, these analyses may also lack statistical power [16,21]. A study on physically inactive adults, which was similar to our target, revealed that the overall effect of intervention on PA was non-significant or minimal [28], suggesting that current interventions are suboptimal. However, no associated BCTs were identified [28], likely due to the low statistical power resulting from the limited number of included studies. Thus, the current research has some limitations restricting its ability to provide more generalizable findings. This exploratory systematic review and meta-analysis aimed to evaluate the effectiveness of health behavior interventions and identify the effective components for improving PA in healthy young and middle-aged adults. The BCT taxonomy enables the accurate extraction of independent and discrete intervention information for the analysis, as well as easily replicable results; therefore, the results of this research can fill the current gap in validating the effectiveness of the interventions and provide an important foundational reference for designing interventions to promote PA among a broad population of healthy young and middle-aged adults.

## 2. Materials and Methods

This systematic review and meta-analysis follow the Preferred Reporting Items for Systematic reviews and Meta-Analyses (PRISMA) statement [29]. The protocol was initially registered in the International Prospective Register of Systematic Reviews (PROSPERO) on 30 April 2022, and revised on 30 October 2024, to include BCT extraction to explore effective components (CRD42022322275).

### 2.1. Study Selection and Eligibility Criteria

We conducted a literature search in two databases (PubMed and Ichu-shi Web). The search queries used for each database are presented in Appendix A. Appendix A shows the meanings of the Japanese keywords used in the Ichu-shi Web database in English. K.K., an expert with extensive experience in developing guidelines for systematic reviews, made major contributions to the design of the search queries. Briefly, the selection criteria focused on randomized controlled trials (RCTs), published before 31 May 2024, that involved exercise or PA interventions targeting healthy young or middle-aged adults. Two independent reviewers (J.W., J.K., T.T., and R.M.) screened studies based on titles and abstracts, followed by a full-text screening when eligible. A study progressed to the next step only if both reviewers agreed; disagreements were resolved by discussing with each other. If the agreements were still not reached, a third reviewer, or a fourth reviewer if it was necessary, was consulted for the discussion (Y.S. or Y.N.). The exclusion criteria involved the following: (1) the article was not a peer-reviewed, original article; (2) the article was not written in English or Japanese; (3) the article was not an RCT; (4) the participants were not humans; (5) the mean age of the participants was <18 years or ≥65 years; (6) the participants included individuals with a disease (such as cancer, myocardial infarction, or diabetes) or special populations (such as pregnant women and military personnel); (7) the intervention was not relative to exercise or PA; (8) the outcome was not exercising habits, PA, or sedentary behavior measured by a device or self-reporting; (9) the intervention period was <4 weeks; (10) the outcome was not presented as the mean changes between the pre- and post-measures or was lacking the standard deviation (SD), standard error (SE), 95% confidence interval (CI), or sample size; and (11) other reasons to be considered ineligible by the reviewers.

### 2.2. Quality Assessment

A quality assessment was conducted according to the Minds Guide for Developing Clinical Practice Guidelines 2020 (Minds guidelines) [30] and the guidelines for systematic reviews and meta-analyses based on the Cochran Handbook [31]. At least two reviewers (J.W., J.K., T.T., R.M., Y.S., or Y.N.) evaluated the research based on the following seven risk-of-bias assessment criteria: (1) *random sequence generation*; (2) *allocation concealment*; (3) the *blinding of the participants and personnel*; (4) the *blinding of the outcome assessment*; (5) *incomplete outcome data*; (6) *selective reporting*; and (7) *other biases*. Each criterion was categorized into “no risk”, “low risk”, or “high risk”. According to the Minds guidelines, the overall bias was assessed qualitatively, rather than quantitatively, based on the level and the number of each risk-of-bias assessment criterion (for example, if the most prevalent risk-of-bias assessment criteria were of a low risk, the overall bias was judged as being a low bias). Each study was initially assessed by one reviewer (J.W., J.K., T.T., or R.M.), and the assessment results were confirmed by another reviewer (J.W., J.K., T.T., or R.M.). If there was a disagreement between the two reviewers, the result was finalized only after a discussion led to a consensus. If the disagreement could not be resolved, a third or fourth reviewer (Y.S. or Y.N.) was consulted.

### 2.3. Data Extraction

As the expert, K.K. delivered professional lectures to other researchers on data extraction, the contents of which are shown below. All the researchers were provided with the Minds guidelines [30] and BCT taxonomy materials [17] and were required to master them before conducting this study. No assessment for training in data extraction, BCT coding, or inter-rater reliability was conducted. J.W., J.K., T.T., and R.M. conducted the data extraction and BCT coding, which were confirmed by another reviewer (J.W. or J.K.) who was not involved in the initial extraction. When two reviewers had different decisions on the extracted data, disagreements were resolved by discussing with a third reviewer or a fourth reviewer (Y.S. or Y.N.) until a consensus was reached. The mean change in the outcomes and their SDs were extracted from each study. If the SD was not provided, Equation (1) was used to convert the SE; for studies with a 95% CI, Equation (2) was used if the group size was ≤100, or Equation (3) was used if the sample size was >100 [31]. We extracted data on total PA, light PA, moderate PA, vigorous PA, moderate and vigorous PA (MVPA), walking, sedentary behavior, and exercise behavior, along with their units, measurement method (device or self-reporting), and domain (overall, work-related, or leisure-related). Additionally, the countries, study designs, intervention periods, intervention methods, BCTs used, group sample sizes, and participants’ ages were extracted for the analysis or to describe the characteristics of included studies.
(1)$$SD=SE\times \sqrt{N}$$
(2)$$SD=\frac{\sqrt{N}\times \left(95\%CI_{\mathrm{upper}\;\mathrm{limit}}-95\%CI_{\mathrm{lower}\;\mathrm{limit}}\right)}{t_{0.025,N-1}}$$
(3)$$SD=\frac{\sqrt{N}\times \left(95\%CI_{\mathrm{upper}\;\mathrm{limit}}-95\%CI_{\mathrm{lower}\;\mathrm{limit}}\right)}{3.92}$$
where N is the sample size and $t_{0.025,N-1}$ is the t-value for a two-tailed t-distribution with a significance level of 0.05 and N−1 degrees of freedom.

### 2.4. Data Analysis

We defined MVPA, total PA, and walking as “PA metric outcomes” to evaluate the comprehensive effect of intervention methods on promoting PA. For studies reporting multiple outcomes, measurement methods, or units, the PA metric outcome for each study was selected in the following order: MVPA > total PA > walking. For the measurement methods, the order was device > self-reporting; for units, it was energy expenditure > distance > steps > time > frequency. The primary and subgroup analyses follow this prioritization order. For studies with multiple intervention or control groups, we used Equations (4)–(6) to combine groups [31]. As reducing sedentary behavior is beneficial for health, we processed its change value by multiplying it by −1 and described this outcome as “reducing sedentary behavior”.
(4)$$N_{pooled}=N_{1}+N_{2}$$
(5)$$Mean_{pooled}=\frac{N_{1}Mean_{1}+N_{2}Mean_{2}}{N_{1}+N_{2}}$$
(6)$$SD_{pooled}=\sqrt{\frac{\left(N_{1}-1\right)SD_{1}^{2}+\left(N_{2}-1\right)SD_{2}^{2}+\frac{N_{1}N_{2}}{N_{1}+N_{2}}\left(Mean_{1}^{2}+Mean_{2}^{2}-2Mean_{1}Mean_{2}\right)}{N_{1}+N_{2}-1}}$$
where $N_{pooled}$ is the pooled sample size, $Mean_{pooled}$ is the pooled mean value, and $SD_{pooled}$ is the pooled SD in the combining group. For cases with three or more intervention or control groups, we sequentially combined the combined group with the remaining groups.

A random-effects model was applied to conduct meta-analyses. The primary meta-analysis aims to analyze intervention effects on overall PA metric outcomes. The subgroup analysis focused on analyzing the effects on different overall outcomes (MVPA, total PA, walking, reducing sedentary behavior, and exercise behavior), as well as on leisure- and work-related PA outcomes. We conducted the meta-regression using the restricted maximum likelihood method. To reduce Type I errors, we included only the BCTs present in three or more studies in the meta-regression, as conducted in a previous study [32]. The merged groups were included in the meta-regression only if the BCTs remained consistent across the merging. When the number of studies was less than the number of BCTs, meta-regression was impossible. As most of the control groups received some intervention, we adopted the following two strategies for the meta-regression: Strategy 1, an aggressive method that ignored the BCTs in the control groups and used only those in the intervention groups; and Strategy 2, a conservative method that subtracted the control group BCTs from the intervention group BCTs before conducting the meta-regression.

In the power analysis, the model type was set to a random-effects model with high heterogeneity, a probability of Type II error = 20% (test power needed to ≥80%), an expected standardized mean difference (SMD) of 0.2, a number of studies, and both sample sizes in the control and intervention groups.

Sensitivity analyses were conducted separately for the meta-analyses and -regressions. The maximum likelihood meta-regression approach was employed to evaluate how the overall study quality influenced the outcomes in the studies included in each meta-analysis and -regression. Studies with a low- or high-risk quality were excluded from the sensitivity analysis when they significantly impacted the meta-analysis results; however, studies with a high risk were excluded from the sensitivity analyses, regardless of their impact. Egger’s test was used to confirm a publication bias. High heterogeneity is desirable as it enables more robust meta-regression. For analyses with a publication bias, the trim-and-fill method was applied to adjust the results in the sensitivity analysis. If the trim-and-fill method failed to address the publication bias, outliers were removed to reduce the impact of the publication bias. Regarding the sensitivity analyses for the meta-regressions, we assessed the impact of the risk of bias and excluded only studies with a high risk. No additional step was taken to specifically adjust the publication bias in the sensitivity analyses for meta-regressions.

The data analysis was performed using the following packages: meta version 7.0.0 [33] for the meta-analysis and visualization, metafor version 4.6.0 [34] for the meta-regression and trim-and-fill method, and dmetar version 0.1.0 [35] for the power analysis and excluding outliers, in R version 4.3.2 on Mac OS (R Foundation for Statistical Computing, Vienna, Austria, 2023). The effect sizes of the SMD were used to evaluate the effectiveness of the interventions and were categorized as small (≥0.2 and <0.5), medium (≥0.5 and <0.8), or large (≥0.8) [36]. The I^2^ and τ^2^ are indicators of heterogeneity, and the p-value represents the significance of the heterogeneity test. The *p*-value < 0.05 was considered statistically significant.

## 3. Results

### 3.1. Search Results

Figure 1 shows the research flow chart. A total of 4266 records were identified, with 3369 and 857 from PubMed and Ichu-shi Web, respectively. After screening the titles and abstracts, 2318 records were excluded. Following a screening of the full text, an additional 1792 records were excluded. The review finally included 116 studies that met the selection criteria and were included in the analyses [37,38,39,40,41,42,43,44,45,46,47,48,49,50,51,52,53,54,55,56,57,58,59,60,61,62,63,64,65,66,67,68,69,70,71,72,73,74,75,76,77,78,79,80,81,82,83,84,85,86,87,88,89,90,91,92,93,94,95,96,97,98,99,100,101,102,103,104,105,106,107,108,109,110,111,112,113,114,115,116,117,118,119,120,121,122,123,124,125,126,127,128,129,130,131,132,133,134,135,136,137,138,139,140,141,142,143,144,145,146,147,148,149,150,151,152].

### 3.2. Characteristics of Included Studies

Appendix A shows detailed information on each study. RCT was the most frequent study design (k = 100), followed by cluster-RCT (k = 14) and crossover-RCT (k = 2). Geographically, most of the studies were conducted in the United States (k = 45), followed by Australia (k = 22), representing 57.8% of the included studies. The total PA was the most frequently reported outcome. Self-reported measurements (k = 71) were more used than devices (k = 56). Most studies reported the total PA (k = 56), followed by walking (k = 48) and MVPA (k = 37). Figure 2 shows the frequencies of the top 10 most used BCTs in the included studies. The BCTs *1.1 Goal setting (behavior)* (k = 70), *2.3 Self-monitoring of behavior* (k = 66), *3.1 Social support (unspecified)* (k = 66), *4.1 Instruction on how to perform a behavior* (k = 64), and *5.1 Information about health consequences* (k = 56) were the top five most frequently used.

### 3.3. Meta-Analyses and Meta-Regressions

Table 1 and Table 2 summarize the results of the meta-analyses and meta-regressions, along with the subsequent sensitivity analyses, respectively. Appendix A present the forest plot for the results of the main and subgroup analyses. Appendix A shows the complete results of the meta-regressions with the subsequent sensitivity analyses. Appendix A shows the heterogeneity in the meta-analyses and meta-regressions with the subsequent sensitivity analyses.

#### 3.3.1. Primary Analysis

There were 102 studies included in the meta-regression of the PA metric [37,38,39,40,41,42,43,48,49,51,53,54,55,56,57,58,60,61,63,64,65,67,68,70,71,72,73,74,75,76,77,78,79,80,81,82,83,84,85,86,87,88,90,91,92,93,94,95,96,97,98,99,101,102,103,104,105,106,107,108,109,110,111,112,113,114,115,116,117,119,120,121,122,123,124,125,126,127,128,129,130,131,132,133,134,135,136,137,138,139,140,141,142,143,145,146,147,148,149,150,151,152]. The random effects model indicated a minimal effect, with an SMD of 0.35 (95% CI, [0.19; 0.50]; *p* < 0.001). The heterogeneity across the studies was significant (I^2^ = 92%, τ^2^ = 0.55, *p* < 0.01).

In total, 90 studies were included in the meta-regression of the PA metric outcomes [37,38,39,41,42,43,49,51,53,54,55,56,57,60,61,63,64,65,67,68,70,71,73,74,75,76,77,78,79,80,81,82,83,84,85,86,87,88,90,91,92,93,95,96,97,98,99,101,102,103,104,105,106,107,108,109,110,113,114,115,116,117,119,120,121,122,123,124,125,127,128,129,130,133,135,136,137,138,139,140,141,142,143,145,146,147,148,149,150,152]. For Strategy 1, *1.5 Review behavior goal(s)* (β = 1.0, *p* = 0.012) showed a significant positive effect. For Strategy 2, *1.5 Review behavior goal(s)* (β = 0.7, *p* = 0.048) and *3.3 Social support (emotional)* (β = 1.0, *p* = 0.041) showed significant positive effects, whereas *3.2 Social support (practical)* (β = −0.9, *p* = 0.028) indicated a significant negative effect.

#### 3.3.2. MVPA

There were 36 studies included in the meta-regression of MVPA [40,43,49,51,54,55,63,65,70,71,72,77,79,80,85,94,99,101,104,111,112,114,117,120,121,122,124,125,127,128,134,140,143,148,150,151]. The interventions in the included studies showed no effect on MVPA, with an SMD of 0.09 (95% CI, [−0.05; 0.22]; *p* = 0.197). The random-effects model of MVPA indicated a high heterogeneity (*p* < 0.01).

In total, 29 studies were included in the meta-regression of MVPA [43,49,51,54,55,63,65,70,71,77,79,80,85,99,101,104,114,117,120,121,122,124,125,127,128,140,143,148,150]. For Strategy 1, *1.2 Problem solving* (β = −0.7, *p* = 0.039) showed a significant negative effect. For Strategy 2, no relative BCT was confirmed.

#### 3.3.3. Total PA

There were 44 studies included in the meta-regression of total PA [39,41,42,48,53,55,56,58,61,64,67,74,75,76,77,78,79,82,83,84,86,87,88,90,91,92,93,95,96,97,101,106,107,110,111,115,116,123,126,128,135,137,138,139,141,146,148,149]. The interventions in the included studies had significant minimal effects on total PA, with an SMD of 0.40 (95% CI, [0.17; 0.61]; *p* < 0.001). The random-effects model of MVPA showed a high heterogeneity (*p* < 0.01).

In total, 44 studies were included in the meta-regression of total PA [39,41,42,53,55,56,61,64,67,74,75,76,77,78,79,82,83,84,86,87,88,90,91,92,93,95,96,97,101,106,107,110,115,116,123,128,135,137,138,139,141,146,148,149]. For Strategy 1, *2.2 Feedback on behavior* (β = 0.8, *p* = 0.011) showed a significant positive effect. For Strategy 2, no relative BCT was confirmed.

#### 3.3.4. Walking

There were 47 studies included in the meta-regression of walking (k = 47) [37,38,39,43,49,54,55,57,60,65,67,68,70,71,73,74,76,79,81,88,91,95,98,102,103,104,105,108,109,111,113,114,117,119,129,130,131,132,133,136,139,142,145,147,148,151,152]. The interventions in the included studies had a significant, small effect on walking, with an SMD of 0.47 (95% CI, [0.21; 0.74]; *p* < 0.001). The random-effects model of walking showed a high heterogeneity (*p* < 0.01).

In total, 43 studies were included in the meta-regression of walking [37,38,39,43,49,54,55,57,60,65,67,68,70,71,73,74,76,79,81,88,91,95,98,102,103,104,105,108,109,113,114,117,119,129,130,133,136,139,142,145,147,148,152]. For Strategy 1, no relative BCT was confirmed. For Strategy 2, *2.3 Self-monitoring of behavior* (β = 0.6, *p* = 0.021), *3.3 Social support (emotional)* (β = 3.4, *p* < 0.001), and *10.2 Material reward (behavior)* (β = 2.2, *p* = 0.003) showed significant positive effects, whereas *3.2 Social support (practical)* (β = −2.9, *p* < 0.001) indicated a significant negative effect.

#### 3.3.5. Reducing Sedentary Behavior

There were 29 studies included in the meta-regression of reducing sedentary behavior [37,53,54,55,56,57,62,65,67,68,70,75,76,77,79,81,84,85,89,101,109,110,120,122,124,125,127,130,140]. The interventions in the included studies had significant, small effects on reducing sedentary behavior, with an SMD of 0.39 (95% CI, [0.09; 0.68]; *p* = 0.013). The random-effects model of reducing sedentary behavior showed a high heterogeneity (*p* < 0.01).

In total, 29 studies were included in the meta-regression of reducing sedentary behavior [37,53,54,55,56,57,62,65,67,68,70,75,76,77,79,81,84,85,89,101,109,110,120,122,124,125,127,130,140]. For Strategy 1, no relative BCT was confirmed. For Strategy 2, *3.1 Social support (unspecified)* (β = 0.9, *p* = 0.003) and *3.3 Social support (emotional)* (β = 2.2, *p* = 0.017) showed a significant positive effect.

#### 3.3.6. Exercise Behavior

There were 7 studies included in the meta-regression of exercise behavior [44,59,74,100,118,139,141]. The interventions in the included studies had a significant effect on exercise behavior, with an SMD of 0.91 (95% CI, [0.13; 1.69]; *p* = 0.029). The random-effects model of exercise behavior showed a high heterogeneity (*p* < 0.01).

As the number of studies was less than the number of BCTs, no subgroup meta-analysis of exercise behavior was conducted for BCTs.

#### 3.3.7. Work-Related PA Metric Outcomes

There were 7 studies included in the meta-regression of the work-related PA metric outcomes (k = 7) [57,81,102,109,125,140,144]. The interventions in the included studies showed no effect on the work-related PA metric outcomes, with an SMD of 0.10 (95% CI, [−0.30; 0.50]; *p* = 0.551). The random-effects model of the work-related PA metric outcomes indicated a high heterogeneity (*p* < 0.01).

As the number of studies was less than the number of BCTs, a subgroup meta-analysis of the work-related PA metric outcomes was not conducted.

#### 3.3.8. Leisure-Related PA Metric Outcomes

There were 12 studies included in the meta-regression of the leisure-related PA metric outcomes [46,47,50,66,69,84,102,111,125,135,141,144]. The interventions in the included studies showed no effect on the leisure-related PA metric outcomes, with an SMD of 0.70 (95% CI, [−0.16; 1.55]; *p* = 0.100). The random-effects model of the leisure-related PA metric outcomes indicated a high heterogeneity (*p* < 0.01).

In total, studies were included in the meta-regression of the leisure-related PA metric outcomes [46,47,50,66,69,84,102,125,135,141,144]. For both Strategies 1 and 2, no relative BCT was confirmed.

### 3.4. Study Qualities and Publication Bias

Figure 3 shows the risk of bias in the 116 included studies [37,38,39,40,41,42,43,44,45,46,47,48,49,50,51,52,53,54,55,56,57,58,59,60,61,62,63,64,65,66,67,68,69,70,71,72,73,74,75,76,77,78,79,80,81,82,83,84,85,86,87,88,89,90,91,92,93,94,95,96,97,98,99,100,101,102,103,104,105,106,107,108,109,110,111,112,113,114,115,116,117,118,119,120,121,122,123,124,125,126,127,128,129,130,131,132,133,134,135,136,137,138,139,140,141,142,143,144,145,146,147,148,149,150,151,152], and Appendix A presents the risk of bias in each included study. Most studies had no overall bias (62.1%). Regarding specific criteria, *random sequence generation* (87.1%) and *other biases* (94.8%) were assess as being of no risk in most of the studies. In contrast, many studies had a low risk of the *blinding of the outcome assessment* (67.2%), *allocation concealment* (57.8%), and *selective reporting* (47.4%). A high risk of *the blinding of the participants and personnel* (62.9%) was present in several studies. Approximately half of the studies had no risk of *incomplete outcome data* (51.6%). The analysis of the impact of the quality of the studies included in the primary and subgroup analyses revealed significant impacts in the meta-analysis and -regression for reducing sedentary behavior (both p of a low risk = 0.044 due to the same included studies). Appendix A shows the impact of study quality on these analyses.

Appendix A shows the publication bias in each analysis, and Appendix A present the funnel plots of each analysis for the different PA outcomes. Egger’s test revealed a significant publication bias in the studies included in meta-analyses on the PA metric outcomes (*p* = 0.017), walking (*p* = 0.012), and the leisure-related PA metric outcomes (*p* = 0.026) among the eight meta-analyses. Although the meta-analyses of exercise behavior (*p* = 0.684) and the work-related PA metric outcomes (*p* = 0.602) did not indicate a significant publication bias, this may be attributed to the limited number of included studies (k < 10), which reduces the statistical power. The meta-regression of walking (*p* = 0.012) and the leisure-related PA metric outcomes (*p* = 0.035) showed a significant publication bias among the six meta-regressions.

### 3.5. Sensitivity Analyses

In the sensitivity analysis for the meta-analyses, the SMD decreased in nearly all the analyses. The meta-analyses for the PA metric outcomes (*p* < 0.001) and total PA (*p* = 0.004) remained effective; however, the SMD of the PA metric outcomes fell below a clinical significance (SMD = 0.19 < 2.0). The meta-analyses on walking (*p* = 0.465), reducing sedentary behavior (*p* = 0.098), and exercise behavior (*p* = 0.250) did not indicate effectiveness, while the analysis of the leisure-related PA metric outcomes showed significant effectiveness in the sensitivity analysis (*p* = 0.016). Appendix A show the forest plots of the results of the main and subgroup analyses.

In the sensitivity analysis for the meta-regression, the results of Strategy 1, 1.5 Review behavior goal(s) for PA metric outcomes (β = 1.1, *p* = 0.012) and 2.2 Feedback on behavior for total PA (β = 1.0, *p* = 0.002) showed a positive effect, while that of 1.2 Problem solving for MVPA indicated a negative effect (β = −0.9, *p* = 0.005). The results of Strategy 2, 2.3 Self-monitoring of behavior (β = 0.8, *p* = 0.033) and 3.3 Social support (emotional) (β = 2.6, *p* = 0.005) for walking showed positive effects, while that of 3.2 Social support (practical) for PA metric outcomes indicated a negative effect. Notably, in the sensitivity analysis, 7.1 Prompts/cues for MVPA showed a negative effect (β = −0.5, *p* = 0.040), while 8.3 Habit formation for total PA exhibited a positive effect (β = 0.7, *p* = 0.012). Although 1.5 Review behavior goal(s) (β = 0.7, *p* = 0.055) and 3.3 Social support (emotional) (β = 1.0, *p* = 0.054) for PA metric outcomes did not show statistical significance in the sensitivity analysis of Strategy 2, the small p-values indicated a potential trend of association.

#### Power Analysis

The power analyses showed that in all the meta-analyses, -regressions, and subsequent sensitivity analyses, the test power was 100%, indicating that the sample size and the number of studies were sufficient for each analysis.

## 4. Discussion

### 4.1. Effectiveness of PA Interventions

This research evaluated the effectiveness of health behavior interventions and identified the effective components for improving PA in healthy adults. In total, 116 studies were included, with 102 used in the primary analysis. The primary analysis results showed that the interventions in the included studies generally increased PA by a small margin, which aligns with the findings of other studies [28]. The subgroup analyses indicated that the interventions in the included studies promoted the overall total PA, reduced sedentary behavior, and increased walking and exercise behavior, with walking showing moderate improvement and exercise behavior demonstrating a large effect. However, in the sensitivity analyses, the interventions showed positive effects only on the PA metric outcomes and MVPA, with a clinically negligible effect on the PA metric outcomes. Although the sensitivity analysis suggested a positive effect on the leisure-related PA metric outcomes, the small number of included studies raises concerns about the reliability of this finding. Nonetheless, the findings highlight an important reality: the clinical effectiveness of current interventions for increasing PA levels remains modest.

### 4.2. Effectiveness of PA Intervention Components

Since its publication in 2013, the BCT taxonomy has been widely used in intervention design [153,154,155,156,157,158]. As most studies include multiple BCTs, verifying the effectiveness of each BCT is challenging. In this research, we used a multivariate meta-regression to control for the influence of other BCTs among healthy young and middle-aged adults. In most studies, the control groups received some level of intervention; therefore, two strategies were employed to comprehensively evaluate the effects of the BCTs. Considering that both strategies led to underestimation and overestimation, BCTs with significant effects in only one strategy, as well as in sensitivity, were considered potential positive or negative components. If they were significant in both strategies, as well as in sensitivity, they were defined as positive or negative components. Based on the results of the primary and sensitivity analyses, 3.2 Social support (practical) was a potential negative component. Since 3.3 Social support (emotional) was significant in the meta-regression and sensitivity analysis for walking by Strategy 2, with a significant trend in the meta-regression for the PA metric outcomes by Strategy 2, it was considered a potentially effective component for the PA metric outcomes in this study. The subgroup and sensitivity analyses highlighted 2.2 Feedback on behavior for the total PA, 2.3 Self-monitoring of behavior for walking, and 3.3 Social support (emotional) for walking as potential positive components, whereas 3.2 Social support (practical) for walking emerged as the potential negative component. Additionally, in the sensitivity analyses of the subgroup analyses, it was found that 7.1 Prompts/cues had a negative effect on MVPA, while 8.3 Habit formation for the total PA emerged as a significant positive effect in both Strategies 1 and 2. The BCTs 3.2 Social support (practical) and 3.3 Social support (emotional) were identified as potential components in this research; however, their effects in the primary and subgroup analyses indicated an association with the detrimental or improved effects on PA.

Currently, limited research exists on how BCTs lead to behavior change. Carey et al. explored the relationship between BCTs and behavior change using mechanisms of actions (MoAs) [159]. Among the results of this research, 2.2 Feedback on behavior is related to Subjective norms and Knowledge, 2.3 Self-monitoring of behavior is related to Behavioral regulation, 8.3 Habit formation is related to Behavioral cueing and Behavioral regulation, 1.2 Problem solving is relevant to Beliefs about capabilities and Environmental context and resources, 3.2 Social support (practical) is related to Social influences and Environmental context and resources, and 7.1 Prompts/cues involves Memory, attention, and decision processes, as well as Behavioral cueing and Environmental context and resources. The BCTs 1.5 Review behavior goal(s) and 3.3 Social support (emotional) were not identified as being related to any MoAs. The BCT 1.5 Review behavior goal(s) is likely related to Goal, and 3.3 Social support (emotional) is likely associated with Feedback processes, while there remains a non-significant association with these MoAs due to the lack of relevant studies. Notably, this study explored the mechanism-related associations only in the frequency of BCTs. Michie et al. linked BCTs according to the Theoretical Domains Framework (TDF) [160], an integrated framework that covers multiple theories [161]. According to the results, 1.5 Review behavior goal(s) is related to Goals, 2.2 Feedback on behavior is related to Knowledge, 2.3 Self-monitoring of behavior is related to Behavioural regulation, 3.3 Social support (emotional) and 3.2 Social support (practical) are associated with Social support or encouragement (general), 8.3 Habit formation is related to Behavioural rehearsal/practice, and 7.1 Prompts/cue is linked to Restructuring the physical environment. The BCT 1.2 problem-solving does not belong to any domain. Linking BCTs to the TDF can enhance the understanding and application; however, due to the immaturity of the theory (with some BCTs unlinked), our findings revealed that, while both BCTs linked to Social support or encouragement (general), emotional support and practical support produced opposing effects. The current theoretical mechanisms do not adequately explain these phenomena in the context of PA.

The BCTs 1.5 Review behavior goal(s), 2.2 Feedback on behavior, and 2.3 Self-monitoring of behavior have been proven in a study previous to the development of the BCT taxonomy and were considered the effective components for promoting self-regulation [162]. It can be presumed that by reviewing behavioral goals, receiving feedback on the current PA behavior, or engaging in self-monitoring, participants gain a clearer understanding of their current status and actively adjust their self-regulation strategies to improve PA levels. This result suggests the feasibility of improving self-regulation to increase PA levels among the general young and middle-aged population. It has been shown that higher levels of emotional social support are associated with increased PA [163,164]. As a mechanism of 3.3 Social support (emotional), fostering close social relationships generates positive role identity and feelings of being useful, which can enhance motivation to maintain a physically active lifestyle [163]. A meta-analysis reported that habit formation interventions are effective in promoting PA [27]. However, interestingly, as an intervention component, 8.3 Habit formation was found to be unrelated to the effects of habit formation interventions. This could be due to 8.3 Habit formation being defined as “prompt rehearsal and repetition of the behavior in the same context repeatedly so that the context elicits the behavior”, while “habit formation” is treated as a composite intervention approach. As its name implies, habit formation enables individuals to engage consistently in PA. Given that 8.3 Habit formation involves repeated practical training or exercises and that 13 of the 23 intervention groups employed self-reported measurements, there is a potential risk that the effectiveness of 8.3 Habit formation was overestimated.

In a meta-analysis of 204 studies on PA self-efficacy, 3.2 Social support (practical) was not a substantial intervention component, as studies excluding it showed higher intervention effects, consistent with our results [165]. Regarding self-efficacy, this phenomenon was explained by 3.2 Social support (practical) potentially reducing their feelings of competence due to the actual assistance from others [165]. The negative effect of 1.2 problem solving shown in this research may be due to similar reason. And we propose another hypothesis that practical support could impose unnecessary burdens on the general adult population, leading to unfavorable effects of the intervention. Additionally, among the 23 intervention groups employing 3.2 Social support (practical), eight involved exercise programs, with six measuring PA using devices. This may be attributed to (1) PA during training sessions might be difficult for devices to capture [166] and (2) fatigue from exercise may lead participants to either avoid daily activities or reduce them due to exhaustion. Either possibility could result in lower measured PA levels in the intervention group. In this study, 7.1 Prompts/cues, primarily implemented as message prompts, exhibited negative effects. Although it was reported that message prompts could enhance PA levels [153,167], a meta-analysis indicated that their overall effect was weak [168]. One possible reason for the diminished effect is message fatigue, which leads to both reactance (active resistance) and inattention (passive resistance) [169]. The finding in this research suggests that healthy young and middle-aged adults are more susceptible to message fatigue, and the excessive use of BCTs could result in adverse effects.

The impact of intervention components on different PA outcomes varied in this study. The effectiveness of increasing step counts using self-monitoring devices has been confirmed [170], which supports the effectiveness of 2.3 Self-monitoring for walking. A study has indicated that a lack of confidence in PA contributes to physical inactivity [171]. Since walking is simpler and more straightforward than other types of PA, a certain degree of emotional support may help reduce psychological barriers, making it easier for individuals to engage in walking. Although MVPA is a subset of total PA, it includes a higher proportion of exercise PA, while total PA encompasses more non-exercise PA [172]. This variation can be considered a reason for the difference between each PA outcome, and we hypothesize that healthy young and middle-aged adults may be more likely to increase PA levels through non-exercise PA rather than exercise PA. However, according to the sensitivity analyses of meta-analyses, the intervention effects for walking and MVPA were not significant. This result may also be influenced by the analytical methods used. Due to the quality of the studies or publication bias, the actual effects of these related BCTs may be smaller, or the negative effects of some BCTs may actually be larger. PA metric outcomes are the comprehensive results of MVPA, total PA, and walking, representing the overall effect of interventions on PA levels. As the prioritization of the three outcomes, the analysis results were influenced by the weighting of each PA outcome. For the differences in studies used for each PA outcome analysis, it is challenging to objectively assess these discrepancies. Clearly, understanding these differences will better inform the design of PA interventions tailored to specific goals, which is a topic for the future.

Intervention programs based on BCTs effectively improved PA levels [28,173,174]. However, as observed in previous studies, not all BCTs contribute positively to increasing PA [16,21,28]. Unfortunately, the theoretical understanding of how BCTs promote PA remains inadequate; thus, further discussion would lack factual support. Moreover, this study aims to identify effective BCTs rather than explore their mechanisms of action. Consequently, this study cannot provide further insights into why and how those relative BCTs influence PA.

### 4.3. Comparisons with Other Similar Studies

Since our research focused on young and middle-aged adults, we compared studies overlapping with these age groups, without restrictions on specific methods or intervention formats. In this research, 1.1 Goal setting (behavior), 2.3 Self-monitoring of behavior, 3.1 Social support (unspecified), 4.1 Instruction on how to perform a behavior, and 5.1 Information about health consequences were the most frequently used components in the included studies. In the 19 studies targeting overweight and obesity during pregnancy, the most frequently used components were 2.3 Self-monitoring of behavior (k=13, 76.5%), 4.1 Instruction on how to perform behavior (k = 13, 76.5%), 5.1 Information about health consequence (k = 8, 47.1%), 1.3 Goal setting (outcome) (k = 7, 41.2%), and 3.1 Social Support (Unspecified) (k = 7, 41.2%) [24]. In the 17 studies focused on employees, the common components were 1.1 Goal setting (behavior) (k = 9, 52.9%), 2.2 Feedback on behavior (k = 9, 52.9%), 12.5 Adding objects to the environment (k = 9, 52.9%), and 5.1 Information about health consequences (k = 7, 41.2%) [25]. Among the 66 studies targeting young adults, the most frequently used components were 4.1 Instruction on how to perform the behavior (k = 40, 47.6%), 1.1 Goal setting (behavior) (k = 38, 45.2%), 1.4 Action planning (k = 36, 42.9%), 2.2 Feedback on behavior (k = 32, 38.1%), and 2.3 Self-monitoring of behavior (k = 31, 36.9%) [23]. Compared to these results, the frequency of BCT usage in our research is similar, reflecting that the methods used in our included studies are comparable to those of previous research. However, as noted above, the frequent use of intervention components does not necessarily indicate greater effectiveness, as such a high frequency may be attributed to researchers’ preferences or personal perspectives. Although the study targeting young adults analyzed the intervention effects of studies using single BCTs, the observed effects might have resulted from other BCTs included in the studies or the combined effects of multiple BCTs. From a methodological perspective, the results derived from our quantitative analysis using meta-regression are more reliable.

However, frequently used intervention components are not always effective. Among the five most frequently used intervention components in the included studies, only 2.2 Feedback on behavior showed potential effectiveness. Compared to a study targeting overweight and obese populations [21], 2.3 Self-monitoring of behavior was identified as a negative intervention through a univariate meta-regression, which is contrary to the results of this study; however, it was not significant in the multivariate meta-regression. Compared to another study targeting overweight and obese populations [16], 2.3 Self-monitoring of behavior was identified as an effective intervention through a univariate meta-regression, showing consistent results with this research; however, it was not significant in the multivariate meta-regression. In a study targeting all physically inactive adults [28], 1.2 Problem solving and 7.1 Prompts/cues were identified as negative intervention components, while 2.3 Self-monitoring of behavior was positive, consistent with our findings. Conversely, 1.5 Review behavior goal(s) and 2.2 Feedback on behavior were found to be negative, contrary to our results. We have not yet confirmed the impact of 3.3 Social support (emotional) and 8.3 Habit formation on PA levels in previous studies. Since the populations in these studies are specific, it is difficult to confirm whether the other BCTs identified as effective can be applied to the general population of healthy young and middle-aged adults. The consistency of results suggests a degree of generalizability, indicating that incorporating these findings into future PA promotion interventions for healthy young and middle-aged adults could enhance their effectiveness.

### 4.4. The Quality of the Evidence

We assessed the impact of study quality on the analysis results and found no significant effect for most analyses, except for those on reducing sedentary behavior, where a significant impact was observed. Nevertheless, only 60% of the included studies had no risk, and excluding all the studies with a risk would likely result in a lower statistical power. Therefore, in the sensitivity analysis, we conservatively excluded only studies with a high risk, unless the study quality significantly influenced the results (the analysis of reducing sedentary behavior). The meta-analyses and -regressions in this research faced challenges due to a publication bias. While the sensitivity analyses of the meta-analyses employing the trim-and-fill method or excluding outliers adjusted the results, they also led to the loss of information from several included studies. By excluding studies with a high risk (for reducing sedentary behavior, studies with a low or a high risk), the sensitivity analysis of the four meta-regressions showed no publication bias. However, the sensitivity analyses of the meta-regressions for walking and the leisure-related PA metric outcomes did not eliminate the publication bias. Egger’s test is a widely used method for detecting a publication bias and is recommended by Cochrane [31]. Conversely, it has been noted to produce false positives when the heterogeneity is high [175]. Ultimately, all the analyses were influenced by the potential quality of the studies. The sensitivity analyses of the meta-analysis results were relatively robust, while the meta-regression results for walking and the leisure-related PA metric outcomes may be more affected by publication bias.

The high heterogeneity among the included studies was identified in most of the analyses (Appendix A), which increased the CI width in the random-effects models. This heterogeneity may be attributed to several factors, including the study duration, basic theory, and sample sizes; combining different intervention components is an important source of heterogeneity [16,21]. Nevertheless, in the subgroup and subsequent sensitivity analyses, the heterogeneity was high. In the meta-regression and the subsequent sensitivity analyses, the models showed a low coefficient of determination, indicating a weak explanatory power for heterogeneity (Appendix A). This suggested that the heterogeneity was not mainly due to the type of PA or BCTs.

### 4.5. Strengths and Limitations

There are some strengths in this research. (1) Currently, research exploring the effectiveness of intervention components for early prevention by promoting PA is limited; however, this research provides new insights into this field; (2) compared with other studies, this research included a larger sample size on which to perform the multivariate meta-regression. We focused on intervention components, rather than specific outcomes, by integrating multiple outcomes into a composite, “PA metric outcomes”; (3) many similar studies did not address how interventions in the control groups were treated. The studies included in this research mostly involved control groups with intervention components. The two strategies used in the meta-regression provided a more comprehensive analysis result. Fourth, some studies included in our research had two or more intervention or control groups. Departing from similar studies, we explicitly stated and combined multiple groups according to recommended methods, which enhanced the credibility of our analytical approach.

This study had some limitations: (1) The use of only two databases was the major limitation of this research, which limited the ability to comprehensively cover relevant studies, as well as the exclusion of all studies with a risk identified during the sensitivity analyses. Given that the results of the exploration are observable, future research should incorporate more databases to enhance comprehensiveness. (2) The use of fewer keywords induced incomplete strategies, which resulted in missing relevant studies. (3) Although we applied the trim-and-fill method or removed outliers to adjust the results of the meta-analyses, there is currently no effective solution to adjust the publication bias in the meta-regression. The right-skewed publication bias may have inflated the perceived effectiveness of certain intervention components. (4) The lack of assessment of the training on data and BCT coding reduced the accuracy of the extractions. The lack of an inter-rater reliability assessment weakens the reproducibility of this study. (5) As most of the studies did not report BCTs, extracting them from text descriptions would introduce human error, which has been reported in other studies [176,177,178]. (6) BCTs can be qualified but not quantified, such as when the intervention group provides health information more frequently than the control group, which may lead to missing BCT groupings in Strategy 2. Future should aim to expand the database searches and establish standardized control group criteria to improve the accuracy of the meta-regression. (7) The lack of timely updates in PROSPERO regarding revision to the research methods has reduced the credibility of this research. (8) We only analyzed changes in PA levels after the intervention without exploring the long-term maintenance effects. (9) The limited number of studies in the subgroup analyses for the leisure-related PA metric outcomes or the absence of a meta-regression for exercise behavior and the work-related PA metric outcomes may result in the inability to effectively assess the association of BCTs. (10) A publication bias and the impact of study quality reduced the accuracy of the analysis results. (11) The high level of unexplained heterogeneity suggests that the discussion of intervention effects in this study is insufficient.

### 4.6. Practical Implications and Recommendations

These study findings have important implications. First, as there was limited discussion on which intervention components are effective in increasing the general health of young and middle-aged adults’ PA, the findings of this study addressed the gap in understanding effective intervention components for promoting PA in this population. Second, the results of the meta-analyses demonstrate the low effectiveness of current interventions, suggesting the need for the continued improvement of most intervention methods. Third, the quantitative meta-regressions of intervention components’ associations offered reliable references for designing intervention methods. Fourth, this study’s findings suggest that healthy young and middle-aged adults tend to favor intervention programs that provide emotional support, self-management, and respect for autonomy.

As the recommendation for practice, interventions focusing on emotional support, autonomy, or promoting self-regulation while avoiding excessive practical support or message reminders may lead to better effects. The conclusion of a qualitative synthesis on adults’ perspectives on maintaining PA includes factors such as physical and mental health, goals, self-monitoring, habits, and accomplishment [179], which indirectly support the practical feasibility of the findings in this study. Regarding PA types, it appears that young and middle-aged adults are more likely to increase their PA levels through daily activities. Intervention components were performed in different forms, including conducting 2.3 Self-monitoring of behavior by subjectively recording exercise data [40], checking accelerometer logs [104], or tracking activities via web-based application [128]. This study’s results can only offer insights from the perspective of the intervention components, rather than more detailed specific methods of implementation. Given the exploratory nature of this study, there are methodological limitations, and the practical applicability is limited. We encourage small-scale confirmatory experiments to validate the findings of this study. However, researchers should interpret the results cautiously and incorporate their professional expertise when designing large-scale formal intervention programs.

### 4.7. Future Directions

As an exploratory study, this research provides a preliminary analysis of the intervention effects and the effectiveness of intervention components in promoting PA among healthy young and middle-aged adults. Improving the methodological limitations recognized in this study is crucial for enhancing the quality of future research. To explore the sources of heterogeneity, previous studies have highlighted the importance of the follow-up duration [27,28,165,176], as well as including other factors related to heterogeneity, such as the number of unique BCTs in the intervention group [16] and the total number of BCTs [28]. As it was reported, these findings show differences compared to those in previous studies [21]. Compared to similar studies conducted with different populations, our research provides generalizable results and reveals that the effects of these BCTs may vary depending on the target population. Understanding which BCTs are generally effective and which are more influenced by specific population characteristics is crucial. This can help identify the source of heterogeneity and design more effective intervention programs tailored to different groups. It is essential not only to identify highly generalizable BCTs but also to explore factors influencing their effectiveness. There is a need for numerous validation experiments on individual BCTs or factorial experiments designed based on a multiphase optimization strategy, as these can generate more direct and reliable foundational evidence [177,180]. Michie et al. integrated the following five levels of intervention characteristics using the word “Behavior Change Ontology”: target behaviors, BCTs, theoretical mechanisms, modes of delivery, and contexts [161]. This research explored the effects of BCTs on the target behavior (PA) within a specific context (healthy young and middle-aged adults) but did not examine the theoretical mechanisms and modes of delivery. Therefore, incorporating these two levels in future studies will be critical for developing effective intervention programs to promote PA and better explain the heterogeneity.

## 5. Conclusions

This study offered new insights into effective intervention elements for increasing PA levels in healthy young and middle-aged adults. The study findings provided a foundational theoretical basis for developing interventions and public health policies to improve PA among young and middle-aged adults, as well as exploratory references for guiding future studies. Overall, interventions had a small effect on promoting PA among healthy young and middle-aged adults. The most effective intervention component identified was 1.5 Review behavior goal(s). The BCTs 2.2 Feedback on behavior, 2.3 Self-monitoring, 3.3 Social support (emotional), and 8.3 Habit formation were potential positive components. In contrast, 1.2 Problem solving, 3.2 Social support (practical), and 7.1 Prompts/cues were potential negative components. As the inclusion for practice, interventions focusing on emotional support, autonomy, or promoting self-regulation, while avoiding excessive practical support or message reminders by improving daily activities, may lead to better effects. However, methodological limitations highlight the need for future improvements, such as expanding databases and refining search strategies.

## Figures and Tables

**Figure 1 behavsci-14-01224-f001:**
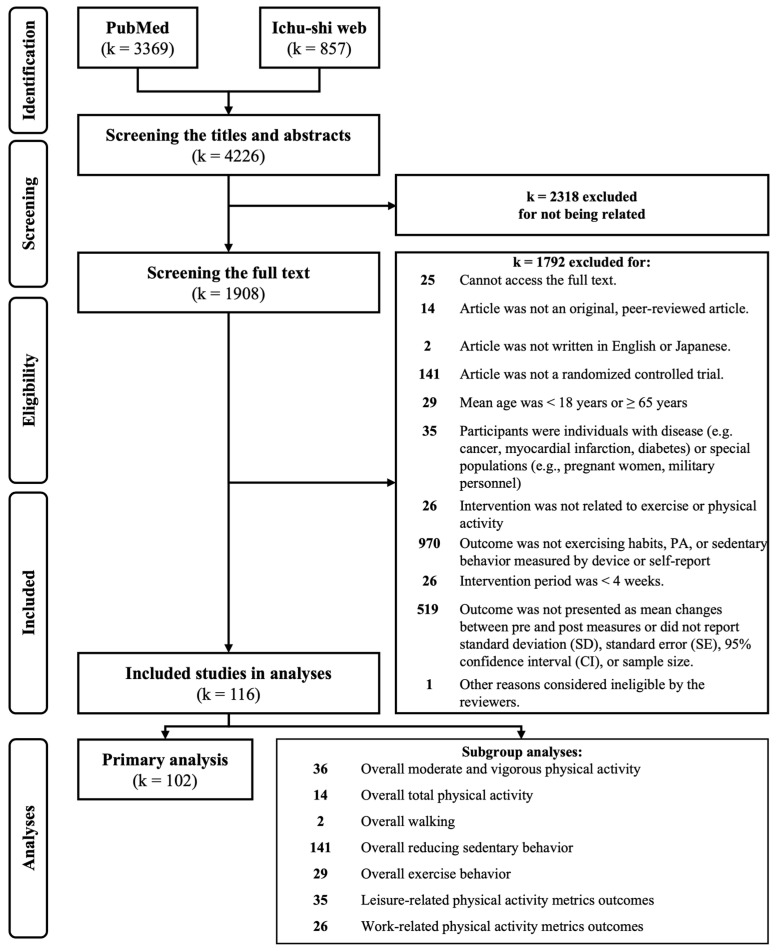
Flowchart of this research. k, numbers of studies. Physical activity metric outcomes were selected in the following order: moderate and vigorous physical activity > total physical activity > walking for each study.

**Figure 2 behavsci-14-01224-f002:**
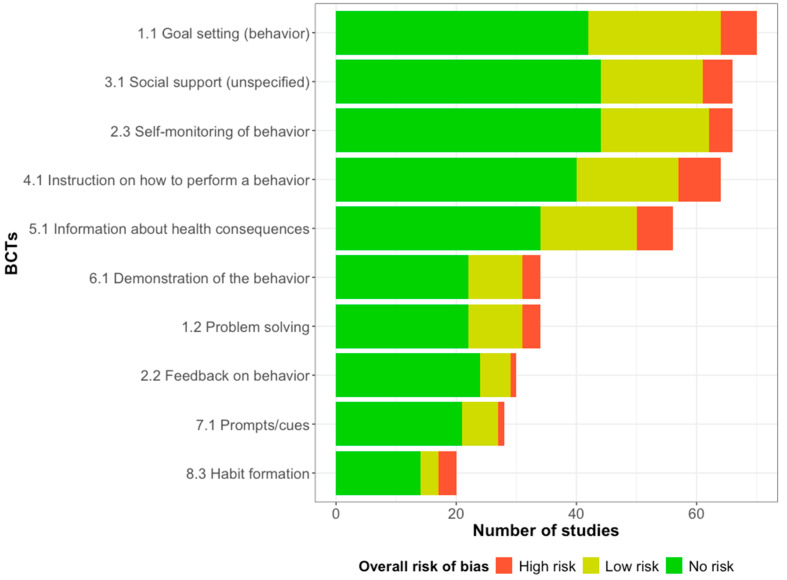
Frequencies of the most used top 10 behavior change techniques (BCTs) in the 116 included studies. The y-axis shows the BCT taxonomy, while the x-axis indicates the number of studies. The BCTs are ranked from top to bottom based on the frequency of use in the included studies. Colors correspond to the overall risk of bias in each study, with green for no risk, yellow for a low risk, and red for a high risk.

**Figure 3 behavsci-14-01224-f003:**
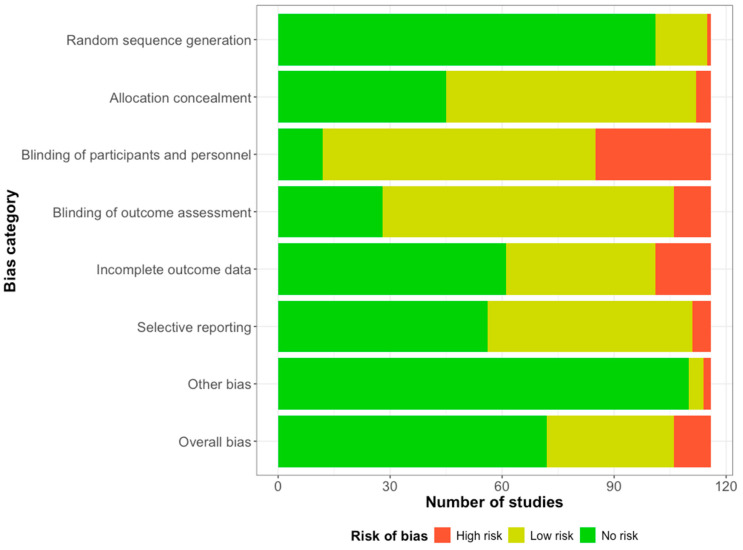
The risk of bias in the 116 included studies. The y-axis shows each bias evaluation item along with the overall bias, and the x-axis indicates the total number of studies. Colors correspond to the risk of bias in each study, with green, yellow, and red representing no, a low, and a high risk, respectively.

**Table 1 behavsci-14-01224-t001:** Result of meta-analyses and sensitivity analyses for primary analysis and subgroup analyses.

Outcome	Meta-Analysis				Sensitivity Analysis			
	k	SMD	95%CI	*p*	k	SMD	95%CI	*p*
Primary analysis								
PA metric outcomes	102	0.35	[0.19; 0.50]	<0.001 *$	73	0.19	[0.13; 0.25]	<0.001 *$
Subgroup analyses								
MVPA	36	0.09	[−0.05; 0.22]	0.197	29	0.08	[−0.07; 0.24]	0.286
Total PA ^a^	48	0.40	[0.17; 0.61]	<0.001 *$	42	0.32	[0.11; 0.53]	0.004 *$
Walking ^b^	47	0.47	[0.21; 0.74]	<0.001 *	43 + 10	0.12	[−0.22; 0.48]	0.465
Reducing sedentary behavior	29	0.39	[0.09; 0.68]	0.013 *	18	0.16	[−0.03; 0.35]	0.098
Exercise behavior	7	0.91	[0.13; 1.69]	0.029 *	4	0.84	[−1.04; 2.72]	0.250
Work-related PA metric outcomes	7	0.10	[−0.30; 0.50]	0.551	7	0.10	[−0.30; 0.50]	0.551
Leisure-related PA metric outcomes ^a^	12	0.70	[−0.16; 1.55]	0.100	9	0.14	[0.03; 0.25]	0.016 *

Note: *, *p* < 0.05; k, numbers of studies; $, significance was confirmed in both meta-analysis and sensitivity analysis; a, removed outliers; b, trimmed and filled, with k shown as number of included studies + number of filled studies; PA, physical activity; MVPA, moderate and vigorous PA; SMD, standardized mean difference; CI, confidence interval. The PA metric outcomes for each study were selected in the following order: MVPA > total PA > walking. Studies with a low risk or a high risk were excluded from the analyses for reducing sedentary behavior, and studies with a high risk were excluded from the analyses for other PA outcomes.

**Table 2 behavsci-14-01224-t002:** The result of meta-regressions and sensitivity analyses for the primary and subgroup analyses, in brief.

Outcome	BCT	Strategy 1										Strategy 2									
		Meta-Regression					Sensitivity Analysis					Meta-Regression					Sensitivity Analysis				
		k	β	SE	P	R^2^	k	β	SE	P	R^2^	k	β	SE	P	R^2^	k	β	SE	P	R^2^
Primary analysis																					
PA metric outcomes	1.5 Review behavior goal(s) ##	10	1.0	0.4	0.012 *$&	0.0%	9	1.1	0.4	0.012 *$	0.0%	9	0.7	0.3	0.048 *&	16.2%	8	0.7	0.4	0.055	18.3%
	3.2 Social support (practical) #	14	−0.5	0.3	0.146		13	−0.5	0.4	0.208		11	−0.9	0.4	0.028 *$		10	−0.8	0.4	0.042 *$	
	3.3 Social support (emotional) #	7	0.7	0.4	0.118		7	0.7	0.5	0.166		6	1.0	0.5	0.041 *		6	1.0	0.5	0.054	
	8.3 Habit formation	16	0.5	0.3	0.127		14	0.6	0.3	0.089		13	0.5	0.3	0.098		11	0.6	0.3	0.041 *	
Subgroup analyses																					
MVPA	1.2 Problem solving #	8	−0.7	0.3	0.039 *$	5.7%	8	−0.9	0.2	0.005 *$	60.2%	7	−0.4	0.3	0.216	0.0%	7	−0.7	0.3	0.056	14.7%
	3.2 Social support (practical)	7	0.4	0.3	0.248		6	0.6	0.3	0.040 *		6	−0.1	0.3	0.744		5	0.5	0.5	0.272	
	7.1 Prompts/cues #	8	−0.4	0.3	0.152		8	−0.5	0.2	0.038 *&		8	−0.4	0.2	0.094		8	−0.5	0.2	0.040 *&	
Total PA	2.2 Feedback on behavior #	10	0.8	0.3	0.014 *$	15.0%	9	1.0	0.3	0.002 *$	32.1%	8	0.6	0.3	0.092	12.3%	7	0.6	0.3	0.078	30.0%
	8.3 Habit formation #	15	0.4	0.3	0.121		13	0.6	0.3	0.021 *&		12	0.4	0.2	0.084		10	0.7	0.2	0.012 *&	
Walking	2.3 Self-monitoring of behavior #	28	0.5	0.4	0.235	8.1%	16	0.5	0.5	0.284	11.0%	22	0.6	0.3	0.021 *$	51.9%	11	0.8	0.4	0.033 *$	39.6%
	3.2 Social support (practical)	7	−1.0	0.6	0.121		4	−1.1	0.7	0.123		4	−2.9	0.7	<0.001 *		2	−1.5	1.1	0.190	
	3.3 Social support (emotional) #	4	1.4	0.7	0.064		3	1.5	0.8	0.073		3	3.4	0.8	<0.001 *$		2	2.6	0.8	0.005 *$	
	10.2 Material reward (behavior)	4	−0.2	0.7	0.781		2	−0.3	0.9	0.700		3	2.2	0.7	0.003 *		1	-	-	-	
Reducing sedentary behavior	3.1 Social support (unspecified)	17	0.0	0.6	0.951	0.0%	16	−0.1	1.2	0.960	0.0%	7	0.9	0.4	0.030 *	0.0%	6	-	-	-	0.0%
	3.3 Social support (emotional)	4	0.3	0.7	0.706		4	-	-	-		4	2.2	0.8	0.017 *		4	-	-	-	

Note: *, *p* < 0.05; k, numbers of studies; R^2^, adjusted coefficient of determination; $, significance was confirmed in both meta-analysis and sensitivity analysis in same meta-regression strategy; &, significance was confirmed in both meta-regression and sensitivity analysis in same analysis; #, was considered as potentially related behavior change technique; ##, was considered as related behavior change technique; PA, physical activity; MVPA, moderate and vigorous PA; Strategy 1, an aggressive method that ignored the BCTs in the control groups and used only those in the intervention groups; Strategy 2, a conservative method that subtracted the control group BCTs from the intervention group BCTs before conducting the meta-regression. The PA metric outcomes for each study were selected in the following order: MVPA > total PA > walking. Studies with a low risk or a high risk were excluded from the analyses for reducing sedentary behavior, and studies with a high risk were excluded from the analyses for other PA outcomes. The restricted maximum likelihood method was used in meta-regressions and the sensitivity analyses of the meta-regressions. Regression analyses were not conducted for exercise behavior and the work-related PA metric outcomes due to the limited number of included studies. No associated BCTs were confirmed in the analysis for leisure-related PA metric outcomes, and the results are not presented in Table 2.

## Data Availability

The data presented in this study are available on request from the corresponding author.

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
