# Peer review of "Effectiveness and Components of Health Behavior Interventions on Increasing Physical Activity Among Healthy Young and Middle-Aged Adults: A Systematic Review with Meta-Analyses"

_behavsci, 2024, doi:10.3390/bs14121224_

Round 1

Reviewer 1 Report

Comments and Suggestions for Authors

1. In line 153, the criteria used in the classification of sizes are indicated: "and were categorized as small (0.2−0.5), medium (0.5−0.8), or large (≥ 0.8)". What exactly did you take into account when the size was 0.5, for example, to consider it small or medium?

2. You intended to follow in this article: the effectiveness of physical activity interventions in order to design future programs aimed at changing the behavior of adults towards the performance of/involvement in physical activities. What were the most effective programs?

3. You have analyzed articles from only two databases. Do you think it is enough or should other articles from other databases also be taken into account? If so, what would they be from your perspective?

Author Response

Comment 1: In line 153, the criteria used in the classification of sizes are indicated: "and were categorized as small (0.2−0.5), medium (0.5−0.8), or large (≥ 0.8)". What exactly did you take into account when the size was 0.5, for example, to consider it small or medium?

Response 1: We thank you for reviewing our manuscript. We apologize for not clearly defining the criteria for effect size. We have revised the description as follows: small (≥ 0.2 and < 0.5), medium (≥ 0.5 and < 0.8), or large (≥ 0.8) [Lines 205–206].

Comment 2: You intended to follow in this article: the effectiveness of physical activity interventions in order to design future programs aimed at changing the behavior of adults towards the performance of/involvement in physical activities. What were the most effective programs?

Response 2: We thank you for your comment. Since our study aims to analyze intervention components, it is difficult to provide a specific intervention program. We have added practical recommendations based on the relevant BCTs identified in 4.6. Practical implications and recommendations in discussion section [Lines 629–636]. However, as intervention components can be performed in various forms in practice, it is difficult to provide more detailed specific practical recommendations [Lines 636–642]:

[Lines 629–636] As the recommendation for practice, interventions focusing on emotional support, autonomy, or promoting self-regulation while avoiding excessive practical support or message reminders may lead to better effects. The conclusion of a qualitative synthesis on adults’ perspectives on maintaining PA includes factors such as physical and mental health, goals, self-monitoring, habits, and accomplishment [176], which indirectly support the practical feasibility of the findings in this study. Regarding PA types, it appears that young and middle-aged adults are more likely to increase their PA levels through daily activities.

[Lines 636–642] Intervention components were performed in different forms, including conducting 2.3 Self-monitoring of behavior by subjectively recording exercise data [40], checking accel-erometer logs [104], or tracking activities via web-based application [128]. This study’s results can only offer insights from the perspective of intervention components rather than more detailed specific methods of implementation. Given the exploratory nature of this study, there are methodological limitations, and the practical applicability is limited.

Comment 3: You have analyzed articles from only two databases. Do you think it is enough or should other articles from other databases also be taken into account? If so, what would they be from your perspective?

Response 3:

We thank you for your kind comment. We think the use of only two databases is the major limitation of this study. About this topic, we revised the discussion on the study limitation to make it clearer [Lines 600–602] as follows:

[Lines 600–602] First, the use of only two databases was the major limitation of this research, which limited the ability to comprehensively cover relevant studies, as well as the exclusion of all studies with risk during sensitivity analyses.

Reviewer 2 Report

Comments and Suggestions for Authors

The title is appropriate and reflects the study content.

The abstract is well-structured and provides essential information.

The novelty/research gap is not adequately justified in relation to similar previous reviews.

Objectives are formulated too generally and lack specific hypotheses.

No clear explanation for specifically choosing young and middle-aged adults.

Lack of contextualization about BCT taxonomy and its validity/reliability.

No justification provided for choosing only two databases (just authors used PubMed and Ichu-shi Web, maybe it is not enough).

Lack of information about reviewer training for data extraction.

The process for resolving discrepancies between reviewers is not detailed.

Risk of bias assessment lacks a standardized protocol.

No explanation for the chosen time period (2020-2024, why not 10 last years? What other similar articles have considered?).

Search strategies are incomplete in supplements.

No prospective protocol registration (registered after the fact).

Lack of sensitivity analysis to evaluate results robustness.

Forest plots are difficult to interpret due to excess information.

Publication bias is not comprehensively analyzed.

Methodological limitations are not adequately discussed.

Lacks critical comparison with similar meta-analyses.

Results interpretation is superficial.

Impact of study quality on results is not analyzed.

Practical implications are vague and general.

No specific recommendations for future research.

Insufficient practical recommendations.

Inconsistencies in abbreviation use.

Tables and figures could be better organized, they have too much information and are too big. In fact, some supplementary data is redundant.

Author Response

Comment 1: The title is appropriate and reflects the study content.

Response 1: We thank you for your reviewing our manuscript and your kind recognition of our title.

Comment 2: The abstract is well-structured and provides essential information.

Response 2: We thank you for your kind recognition of our abstract.

Comment 3: The novelty/research gap is not adequately justified in relation to similar previous reviews.

Response 3: We thank you for your kind comment. We apologize for not outlining the novelty of this study. Due to the current lack of research on young and middle-aged adults (Comment 5), there is an absence of a theoretical basis when designing intervention programs, which poses a risk of poor intervention effectiveness. Given the lack of similar prior research, this study serves as an exploratory investigation, offering an initial analysis of the intervention components that could enhance physical activity. As noted in your other comments, this study has some limitations. However, these exploratory results not only offer some initial theoretical evidence for practice but also facilitate subsequent analysis after refining the methodology. Accordingly, we have described the novelty and significance of this study through a simple comparison with previous research and emphasized the value of this study. [Lines 78–92].

Comment 4: Objectives are formulated too generally and lack specific hypotheses.

Response 4: We thank you for your kind comment. Improving physical activity levels among healthy young and middle-aged adults holds great societal importance (Comment 5). However, the current lack of evaluations on intervention effectiveness and the validity of intervention components may result in insufficient evidence to guide intervention design (Comment 3). It may lead to the selection of ineffective components, ultimately reducing intervention effects. By focusing on a wide-ranging population, this research helps to deliver generalizable evidence to overcome the problem present in the current intervention design. In future research, we will gradually narrow the focus to explore results in more specific subgroups of this population.

Comment 5: No clear explanation for specifically choosing young and middle-aged adults.

Response 5: We thank you for your kind comment. We apologize for not providing a detailed explanation. The number of young and middle-aged adults is large, and they hold great social importance. However, due to their lack of physical activity, they are exposed to high health risks. From a health prevention perspective, increasing their physical activity is crucial. Accordingly, we have added the relevant explanation to the background section of the revised manuscript [Lines 50–61] as follows:

[Lines 50–61] The working-age population comprises approximately 65% of the total population [5], playing a crucial role in societal productivity and development. Within this demographic, young and middle-aged adults form the majority, making efforts to increase their PA to improve health and prevent diseases highly critical for society. A study utilizing data from the National Health and Nutrition Examination Survey reported that 47.0% and 59.8% of individuals aged 18–44 and 45–64 years, respectively, did not meet recom-mended PA levels in the United States [6]. Although some young and middle-aged adults appear healthy, physical inactivity exposes them to high health risks. Thus, increasing PA before health problems emerge is crucial for ensuring sustained health benefits in the early stage and has been widely recognized [7]. Numerous studies exist on promoting PA among young and middle-aged adults [8-13]; however, the evaluation of their overall effectiveness remains limited.

Comment 6: Lack of contextualization about BCT taxonomy and its validity/reliability.

Response 6:

We thank you for your kind comment. We apologize for not providing contextualization of the BCT taxonomy and its validity/reliability. Accordingly, we have added some evidence regarding the effectiveness of BCT in the background [Lines 70–77].

[Lines 70–77] BCT is defined as the observable, replicable, and irreducible component of the intervention designed to alter or redirect causal processes that regulate behavior [17]. The development of the BCT taxonomy was based on published technical theories and utilized a structured decision-support technique known as the Delphi method. Through iterative rounds of opinion collection, integration, reliability testing, and feedback, a systematic taxonomy with high consensus and reliability was achieved [17]. The theoretical application of BCT has been confirmed to effectively promote improvements in various health behaviors, including PA [18-20].

Comment 7: No justification provided for choosing only two databases (just authors used PubMed and Ichu-shi Web, maybe it is not enough).

Response 7: We thank you for your kind comment. We think the use of only two databases is the major limitation of this study. About this topic, we have revised the discussion on the study limitation to make it clearer [Lines 601–60] as follows:

[Lines 601–603] First, the use of only two databases was the major limitation of this research, which limited the ability to comprehensively cover relevant studies, as well as the exclusion of all studies with risk during sensitivity analyses.

Comment 8: Lack of information about reviewer training for data extraction.

Response 8: We thank you for your kind comment. We apologize for the lack of information about reviewer training for data extraction. The researchers received training on data extraction from experts and were provided with relevant reference materials; however, no evaluation of the data extraction training was conducted. Accordingly, we have added details about reviewer training to the method section [Lines 138–141]. Due to the lack of assessment in data extraction, we have also addressed its impact in the study limitations [Lines 609–610].

[Lines 138–141] As the expert, K.K. delivered professional lectures to other researchers on data extraction, the contents of which are shown below. All researchers were provided with the Minds guidelines [30] and BCT taxonomy materials [17] and were required to master them before conducting this study. No assessment for training was conducted.

[Lines 609–610] Fourth, the lack of assessment in training on data and BCT extraction reduced the accuracy of the extractions.

Comment 9: The process for resolving discrepancies between reviewers is not detailed.

Response 9: We thank you for your kind comment. We apologize for the undetailed description. If two reviewers had discrepancies, the two reviewers would first exchange their reasons and attempt to reach an agreement by discussion. If the discrepancies persisted, a third reviewer would be consulted for the discussion. If the agreement was still not reached, a fourth reviewer would join the discussion. Thus, “discussion” was central to resolving the discrepancies; therefore, we have emphasized this approach in the manuscript, with appropriate revisions to the manuscript [Lines 108–111, 134–136, 143–145].

[Lines 108–111] A study progressed to the next step only if both reviewers agreed; disagreements were resolved by discussing with each other. If the agreements were still not reached, a third reviewer, or a fourth reviewer if it was necessary, was consulted for the discus-sion (Y.S. or Y.N.).

[Lines 134–136] As mentioned above, any disagreement was resolved by discussion, and if necessary, the third or fourth reviewer was consulted (Y.S. or Y.N.).

[Lines 143–145] When two reviewers had different decisions on the extracted data, disagreements were resolved by discussing with a third reviewer or a fourth reviewer if it was necessary (Y.S. or Y.N.).

Comment 10: Risk of bias assessment lacks a standardized protocol.

Response 10: We thank you for your kind comment. We apologize for the insufficient description in this regard. The bias was assessed based on the Minds Handbook for Clinical Practice Guideline Development 2020 (only in Japanese, https://minds.jcqhc.or.jp/methods/cpg-development/minds-manual/), and additional details have been included in the revised manuscript [Lines 123–134].

[Lines 123–134] Quality assessment was conducted according to the Minds Guide for Developing Clinical Practice Guidelines 2020 (Minds guidelines) [30] and the guidelines for systematic review and meta-analysis based on the Cochran Handbook [31]. At least two reviewers (J.W., J.K., T.T., R.M., or Y.S.) evaluated the research based on the following seven risk of bias assessment criteria: (1) random sequence generation; (2) allocation concealment; (3) blinding of participants and personnel; (4) blinding of outcome assessment; (5) incomplete outcome data; (6) selective reporting; and (7) other bias. Each criterion was categorized into "no risk," "low risk," or "high risk." According to the handbook, the overall bias was assessed qualitatively rather than quantitatively based on the level and number of each risk of bias assessment criteria (for example, if the most prevalent risks of bias assessment criteria were low risk, the overall bias was judged as low bias). Therefore, the final decision on the overall risk of bias was the result of consensus discussions among at least two reviewers.

Comment 11: No explanation for the chosen time period (2020-2024, why not 10 last years? What other similar articles have considered?).

Response 11: We thank you for your kind comment. We apologize for the ambiguity caused by the description of the study period. The period 2020–2024 refers to the duration of searching articles, not the publication years of the chosen articles. In this study, we included all studies published before the last search date (May 31, 2024). To avoid ambiguity, slight revisions were made in section 2.1 Study selection and eligibility criteria [Line 100, 105-106].

[Line 100]

Old: We conducted a literature search in two databases (PubMed and Ichu-shi Web) from April 2020 to May 2024, following protocol revisions.

New: We conducted a literature search in two databases (PubMed and Ichu-shi Web).

[Lines 105-106]

Old: In brief, the selection criteria focused on randomized controlled trials (RCT) involving exercise or PA interventions targeting healthy young or middle-aged adults.

New: Briefly, the selection criteria focused on randomized controlled trials (RCT) published before May 31, 2024, that involved exercise or PA interventions targeting healthy young or middle-aged adults.

Comment 12: Search strategies are incomplete in supplements.

Response 12: We thank you for your comment. We acknowledge the issue of incompleteness in the search strategy and sincerely apologize for it. The primary contribution to the search strategy’s development was made by K.K., an experienced expert who participated in making several editions of the Minds Handbook for Clinical Practice Guideline Development, which is mentioned in Comment 8. K.K.’s involvement ensures the reliability of the established search strategy. However, as you highlighted, a simple search strategy will result in insufficient coverage of the literature. This limitation has been addressed in our discussion of study limitations [Lines 604–605]. In the future, we will seek advice from multiple experts and refer to other studies to develop more complete search strategies.

[Lines 604–605] Second, the use of fewer keywords induced incomplete strategies, which resulted in missing relevant studies. 

Comment 13: No prospective protocol registration (registered after the fact).

Response 13: We thank you for your kind comment. We apologize for the lack of a detailed statement. The study protocol was first registered in 2020. During data extraction, we incorporated the use of the BCT taxonomy to better analyze intervention components. However, we did not update the PROSPERO registration in a timely manner, which may have affected the credibility of our study. Since the PROSPERO registration page provides historical versions, it could be confirmed that our revisions have minimal deviations from the first version of the protocol. This fact has been addressed in the study limitations [Lines 616–617] as follows:

[Lines 616–617] Seventh, the lack of timely updates in PROSPERO regarding revision to the research methods has reduced the credibility of this research.

Comment 14: Lack of sensitivity analysis to evaluate results robustness.

Response 14: We thank you for your comment. We apologize for not conducting a sensitivity analysis to enhance the robustness of our study results. By referring to the Cochrane guidelines, previous research, and your other comments, we conducted sensitivity analyses for study quality and publication bias. This content has been added to the methods and results sections of the manuscript, and the findings have been discussed. We found that the results of the sensitivity analysis were consistent with most previous findings. We appreciate your suggestion, which contributed to strengthening the robustness of our analysis. Additional sensitivity analysis methods [Lines 188–200], results [Lines 379–400], and discussion of the findings [Lines 410–416, 423–432, 435–438] have been added to the manuscript.

Comment 15: Forest plots are difficult to interpret due to excess information.

Response 15: We thank you for your kind comment. Due to the extensive number of included studies, the forest plot may seem redundant. To improve clarity and reader engagement, we have summarized the results of all meta-analyses into a single table [Table 1].

Comment 16: Publication bias is not comprehensively analyzed.

Response 16: We thank you for your kind comment. We conducted publication bias assessments for all analyses, including subgroup analyses. In the sensitivity analysis of the meta-analysis, we applied the trim and fill method or removed outliers to adjust the results. For the meta-regression, we reviewed some literature but found no reliable method to adjust the publication bias present in the meta-regression. We have described this topic in the results [Lines 364–373] and discussion [Lines 571–585] sections.

Comment 17: Methodological limitations are not adequately discussed.

Response 17: We thank you for your kind comment. We appreciate the valuable comments provided by each reviewer. According to those comments, we have revised the study limitations [Lines 600–619].

Comment 18: Lacks critical comparison with similar meta-analyses.

Response 18: We thank you for your kind comment. We sincerely apologize for not conducting a detailed comparison with previous studies. In 4.3 Comparisons with other similar studies of the discussion section, we compared our results with those of some previous studies with similar methods [Lines 518–562].

Comment 19: Results interpretation is superficial.

Response 19: We thank you for your kind comment. We sincerely apologize for not providing a sufficient explanation of the results. Accordingly, we have structured our discussion into the following two parts: the intervention effects from the meta-analyses [Lines 402–416] and the associations of intervention components identified in the meta-regressions [Lines 417–517]. In the associations of intervention components identified in the meta-regressions, we categorize them into the following four aspects: determining the effectiveness of BCTs [Lines 418–441], explaining the mechanisms of BCTs based on previous theoretical research [Lines 442–456], interpreting the mechanisms of positive and negative BCTs [Lines 457–496] from a practical perspective, and exploring the reasons for the differences in BCTs across different PA outcomes (suggested by Reviewer 3) [Lines 497–517].

Comment 20: Impact of study quality on results is not analyzed.

Response 20: We thank you for your kind comment. As outlined in Comment 14, we have detailed the approach used to assess the impact of study quality. This topic has been further discussed in the discussion section. We used meta-regression to confirm the impact of study quality on each analysis. This content overlaps slightly with comment 14; however, it has been addressed in the methods [Lines 189–193], results [Lines 353–363], and discussion [Lines 564–570] sections.

Comment 21: Practical implications are vague and general.

Response 21: We thank you for your kind comment. We apologize for the practical implications being vague and general. Accordingly, we have described the practical implications in 3.6. Practical implications and recommendations in discussion section [Lines 621–628].

[Lines 622–629] These study findings have important implications. First, as there was limited discussion on which intervention methods are effective in increasing the general health of young and middle-aged adults' PA, the findings of this study addressed the gap in understanding effective intervention components for promoting PA in this population. Second, the results of meta-analyses demonstrate the low effectiveness of current interventions, suggesting the need for continued improvement of most intervention methods. Third, the quantitative meta-regressions of intervention components’ associations offered reliable references for designing intervention methods.

Comment 22: No specific recommendations for future research.

Response 22: We thank you for your kind comment. We apologize for not discussing specific recommendations for future research. This topic has been addressed in 3.8 Future Directions in the discussion section [Lines 643–659].

Comment 23: Insufficient practical recommendations.

Response 23: We thank you for your kind comment. We apologize that our practical recommendations are insufficient. Accordingly, we have added practical recommendations based on the relevant BCTs identified in 4.6. Practical implications and recommendations in discussion section [Lines 629–636]. However, as intervention components can be performed in various forms in practice, it is difficult to provide more detailed specific practical recommendations [Lines 636–642].

[Lines 629–636] As the recommendation for practice, interventions focusing on emotional support, autonomy, or promoting self-regulation while avoiding excessive practical support or message reminders may lead to better effects. The conclusion of a qualitative synthesis on adults’ perspectives on maintaining PA includes factors such as physical and mental health, goals, self-monitoring, habits, and accomplishment [176], which indirectly support the practical feasibility of the findings in this study. Regarding PA types, it appears that young and middle-aged adults are more likely to increase their PA levels through daily activities.

[Lines 636–642] Intervention components were performed in different forms, including conducting 2.3 Self-monitoring of behavior by subjectively recording exercise data [40], checking accelerometer logs [104], or tracking activities via web-based application [128]. This study’s results can only offer insights from the perspective of intervention components rather than more detailed specific methods of implementation. Given the exploratory nature of this study, there are methodological limitations, and the practical applicability is limited.

Comment 24: Inconsistencies in abbreviation use.

Response 24: We thank you for your kind comment. We reviewed the spelling of all abbreviations, including those in the supplementary files, and identified an error in Figure 1, where “Ichu-shi Web” was incorrectly spelled. This has been corrected. The spelling error in “PA metric outcomes” has been identified and corrected, ensuring consistency across the manuscript.

Comment 25: Tables and figures could be better organized, they have too much information and are too big. In fact, some supplementary data is redundant.

Response 25: We thank you for your kind comment. We apologize for presenting the disorganized results, which led to a poor reading experience. As noted in Comment 15, the main data and results of the meta-analysis and -regression are summarized in Tables 1 and 2. Considering the need to review the details of each analysis, we have included all forest plots, funnel plots, and complete regression analysis results in the supplementary materials.

Reviewer 3 Report

Comments and Suggestions for Authors

The manuscript addresses a relevant topic in public health. The use of a rigorous methodological approach (meta-analysis and meta-regression) and the focus on specific intervention components (BCT) are clear strengths. However, the following considerations are raised for the improvement of the quality of the work:

-Regarding the BCT Taxonomy, a more detailed explanation of why these specific techniques (e.g., goal setting, self-monitoring) are relevant to the design of interventions for healthy adults is missing.

-Although the general objective is mentioned, no clear hypotheses or specific research questions are specified. This should be added to better guide the reader.

-It is acknowledged that only two databases were used. It is crucial to emphasise how this choice may have limited the scope of the included studies.

-Although supplementary figures (e.g., S1 to S10) are mentioned, they are not described in detail in the main text. It would be useful to include a brief summary of each figure for readers who do not access the supplement.

-Although subgroup analyses are conducted (e.g., MVPA, walking), there is little discussion of why these results differ between subgroups. It would be useful to propose hypotheses or interpretations based on the data.

-Although similar studies are cited, the manuscript does not contextualise how their findings make a novel contribution to the field.

 -It would be useful to suggest how practitioners might apply these findings to the design of real-world interventions.

Author Response

The manuscript addresses a relevant topic in public health. The use of a rigorous methodological approach (meta-analysis and meta-regression) and the focus on specific intervention components (BCT) are clear strengths. However, the following considerations are raised for the improvement of the quality of the work:

Comment 1: Regarding the BCT Taxonomy, a more detailed explanation of why these specific techniques (e.g., goal setting, self-monitoring) are relevant to the design of interventions for healthy adults is missing.

Response 1: We thank you for reviewing our manuscript and for your kind comment. We introduced the mechanisms by which the relevant BCTs in this study function through a theoretical analysis (BCT and MoA) [Lines 442–455]. Then, we explored why these BCTs are effective from practical perspectives [Lines 457–496]. Some BCTs, such as those with self-explanatory names, were not specifically defined. However, we clarified the definition of 8.3 Habit formation due to potential confusion. All explanations provided were based on the BCT taxonomy.

Comment 2: Although the general objective is mentioned, no clear hypotheses or specific research questions are specified. This should be added to better guide the reader.

Response 2: We thank you for your kind comment. As this was an exploratory study, we could not establish any hypotheses. However, given that the current discussions on this topic for general healthy young and middle-aged adults were still insufficient, we have emphasized this problem with why we focused on healthy young and middle-aged adults [Lines 50–61] and the research gap [Lines 78–92] in the background section.

Comment 3: It is acknowledged that only two databases were used. It is crucial to emphasise how this choice may have limited the scope of the included studies.

Response 3: We thank you for your kind comment. Compared to other systematic reviews, the use of only two databases has limited the scope of our analysis, which led to the major limitation of this study. We plan to use additional databases in future analyses. We have revised the discussion on the study limitation to make it clearer [Lines 600–602].

[Lines 600–602] First, the use of only two databases was the major limitation of this research, which limited the ability to comprehensively cover relevant studies, as well as the exclusion of all studies with risk during sensitivity analyses.

Comment 4: Although supplementary figures (e.g., S1 to S10) are mentioned, they are not described in detail in the main text. It would be useful to include a brief summary of each figure for readers who do not access the supplement.

Response 4: We thank you for your kind comment. We apologize for not providing a better reading experience. To address this, we have added summary tables for meta-analyses, meta-regressions, and their sensitivity analyses to the main text [Tables 1 and 2].

Comment 5: Although subgroup analyses are conducted (e.g., MVPA, walking), there is little discussion of why these results differ between subgroups. It would be useful to propose hypotheses or interpretations based on the data.

Response 5: We thank you for your kind comment. We apologize for not providing sufficient discussion on this aspect. Accordingly, relevant discussions have been added to the discussion section of the manuscript [Lines 497–517].

Comment 6: Although similar studies are cited, the manuscript does not contextualise how their findings make a novel contribution to the field.

Response 6:

We thank you for your kind comment. We briefly introduced some studies in the background section and explained that their findings were limited to specific aspects [Lines 78–85]. In 4.3. Comparisons with other similar studies in the discussion section, we presented their findings [Lines 523–534, 545–556].

Comment 7: It would be useful to suggest how practitioners might apply these findings to the design of real-world interventions.

Response 7: We thank you for your kind comment. Accordingly, we have added practical recommendations based on the relevant BCTs identified in 4.6. Practical implications and recommendations in the discussion section [Lines 629–636]. However, as intervention components can be performed in various forms in practice, it is difficult to provide more detailed specific practical recommendations [Lines 636–642]:

[Lines 629–636] As the recommendation for practice, interventions focusing on emotional support, autonomy, or promoting self-regulation while avoiding excessive practical support or message reminders may lead to better effects. The conclusion of a qualitative synthesis on adults’ perspectives on maintaining PA includes factors such as physical and mental health, goals, self-monitoring, habits, and accomplishment [176], which indirectly support the practical feasibility of the findings in this study. Regarding PA types, it appears that young and middle-aged adults are more likely to increase their PA levels through daily activities.

[Lines 636–642] Intervention components were performed in different forms, including conducting 2.3 Self-monitoring of behavior by subjectively recording exercise data [40], checking accelerometer logs [104], or tracking activities via web-based application [128]. This study’s results can only offer insights from the perspective of intervention components rather than more detailed specific methods of implementation. Given the exploratory nature of this study, there are methodological limitations, and the practical applicability is limited.

Round 2

Reviewer 2 Report

Comments and Suggestions for Authors

Research gap not clearly articulated

Outdated references (some pre-2000)

Limited discussion of previous systematic reviews

Theoretical framework for BCTs needs expansion

Connection between BCTs and physical activity could be better explained

Only two databases searched (PubMed and Ichu-shi Web) - insufficient for systematic review standards

No inter-rater reliability statistics provided

Missing power analysis/sample size calculations

Inadequate description of risk of bias assessment process

Limited explanation of reviewer disagreement resolution

No mention of protocol pre-registration

Training for BCT coding not assessed

Very high heterogeneity (I² = 92%) not adequately addressed

Clear publication bias present but inadequately explored

Some meta-regressions based on too few studies (k < 10)

Wide confidence intervals in several analyses

Insufficient reporting of effect sizes

Missing subgroup analyses details

Inadequate discussion of heterogeneity sources

Some interpretations exceed what the data supports

Clinical significance of effects not adequately discussed

Future research directions too vague

Implementation considerations limited

BCT mechanisms not fully explored

Limited discussion of methodological limitations

No adjustment for multiple comparisons

Publication bias adjustment methods inadequate

Inconsistent citation format

Missing DOIs in references

Author Response

Comment 1: Research gap not clearly articulated

Response 1: We thank you for your kind comment. Accordingly, we have provided a more detailed discussion of the limitations in the current study to clarify the research gap and the novelty in our research [Lines 78–99].

Comment 2: Outdated references (some pre-2000)

Response 2: We thank you for your kind comment. Accordingly, we reviewed the publication dates of the references and made the following adjustments to outdated references or explained why we chose not to update them:

<References 37–152> The studies we included in our search. Our search covered studies published before May 31, 2024; therefore, some outdated studies were also included.

<Reference 7 Harris et al. 1989> We referenced this study to highlight that the concept of “increasing physical activity as a preventive measure” has been acknowledged for a long time.

<Reference 36 Cohen 1988> Cohen’s d as the standard is very classic, widely used in many studies and was cited in the Cochrane Handbook.

[Lines 58–59]

Old: Thus, increasing PA before health problems emerge is crucial for ensuring sustained health benefits in the early stage and has been widely recognized [7].

Revised: Thus, increasing PA before health problems emerge is crucial for ensuring sustained health benefits in the early stage and has been long widely recognized [7].

Comment 3: Limited discussion of previous systematic reviews

Response 3: We thank you for your kind comment. As with other systematic reviews, discussions regarding BCTs typically consider their consistency with previous studies and briefly explore the mechanism of effectiveness. In this study, due to structural considerations, we addressed the impact mechanisms of the findings in 4.2. Effectiveness of PA intervention components [Lines 439–558] and subsequently compared them with other studies in 4.3. Comparisons with other similar studies [Lines 559–604]. The findings revealed the generalizability of our findings; however, for certain populations, the applicability of some BCTs may be limited. Since our research focuses on age, comparisons were only made with studies with age-matched populations. The studies mentioned in the Background section on specific methods or intervention formats inherently constrain the use of BCTs, limiting the scope of coverage. To highlight this, we have revised the following sentences [Lines 560–562]:

[Lines 560–562]

Old: Due to a few similar studies, we compared our findings with those from studies that overlap in age range with young and middle-aged adults.

Revised: Since our research focuses on young and middle-aged adults, we compared studies overlapping with these age groups, without restrictions on specific methods or intervention formats.

Comment 4: Theoretical framework for BCTs needs expansion

Response 4: We thank you for your kind comment. Accordingly, we have used the TDF to explain the theoretical mechanisms of BCTs [Lines 477–489]. As a result of the discussion, based on current theory, it is difficult to make a definitive conclusion about how BCTs influence physical activity.

[Lines 477–489]

Added: Michie et al. linked BCTs according to the Theoretical Domains Framework (TDF) [160], an integrated framework that covers multiple theories [161]. According to the results, 1.5 Review behavior goal(s) is related to Goals, 2.2 Feedback on behavior is related to Knowledge, 2.3 Self-monitoring of behavior is related to Behavioural regulation, 3.3 Social support (emotional) and 3.2 Social support (practical) are associated with Social support or encouragement (general), 8.3 Habit formation is related to Behavioural rehearsal/practice, and 7.1 Prompts/cue is linked to Restructuring the physical environment. The BCT 1.2 problem-solving does not belong to any domain. Linking BCTs to the TDF can enhance the understanding and application; however, due to the immaturity of the theory (with some BCTs unlinked), our findings revealed that, while both BCTs linked to Social support or encouragement (general), emotional support and practical support produced opposing effects. The current theoretical mechanisms do not adequately explain these phenomena in the context of PA.

Comment 5: Connection between BCTs and physical activity could be better explained

Response 5: We thank you for your kind comment. Most previous studies focus on identifying which BCTs influence PA; however, there is a lack of conclusive evidence on how these BCTs affect PA (their investigations frequently involve subjective inferences to varying degrees). The connection between PA and BCTs still requires substantial empirical validation, which we have mentioned as a future research direction. Furthermore, this study does not aim to discuss the mechanisms of BCTs but to explore which BCTs are effective. We can only infer potential reasons for the association of these BCTs based on existing evidence to justify their relevance. Thus, we have included the following statement in the Discussion section [Lines 552–558]:

[Lines 552–558]

Added: Intervention programs based on BCTs effectively improved physical activity levels [28, 173, 174]. However, as observed in previous studies, not all BCTs contribute positively to increasing physical activity [16, 21, 28]. Unfortunately, the theoretical understanding of how BCTs promote physical activity remains inadequate; thus, further discussion would lack factual support. Moreover, this study aims to identify effective BCTs rather than explore their mechanisms of action. Consequently, this study cannot provide further insights into why and how those relative BCTs influence PA.

Comment 6: Only two databases searched (PubMed and Ichu-shi Web) - insufficient for systematic review standards

Response 6: We thank you for your kind comment. As the comment made in the Round 1 revisions, the use of only two databases is one of the main limitations of this study. We have addressed this in the study limitations section of the discussion [Lines 648–650]. As stated in this article, this study serves as an exploratory investigation providing preliminary findings. In the future, we will actively incorporate suggestions from peers to refine our methods and contribute more reliable references to this field.

[Lines 648–650] the use of only two databases was the major limitation of this research, which limited the ability to comprehensively cover relevant studies, as well as the exclusion of all studies with risk during sensitivity analyses.

Comment 7: No inter-rater reliability statistics provided

Response 7: We thank you for your kind comment. We apologize for not providing inter-rater reliability statistics. Since there were no descriptions of inter-rater reliability statistics in the previous studies we referenced, we did not have the relevant records for re-assessment. This problem has been addressed in the Methods [Lines 149–150] and study limitations in the Discussion section [Lines 658–659].

[Lines 149–150]

Old: No assessment for training was conducted.

Revised: No assessment for training in data extraction, BCT coding or inter-rater reliability was conducted.

[Lines 658–659]

Added: The lack of an inter-rater reliability assessment weakens the reproducibility of this study;

Comment 8: Missing power analysis/sample size calculations

Response 8: We thank you for your kind comment. We conducted the power analyses to assess whether the sample size and the number of studies included in each analysis were sufficient to detect an SMD of 0.2. The results showed that all analyses had adequate sample sizes and study numbers. The power analysis has been described in the Methods [Lines 197–200] and Results [Lines 420–423] sections. Since the power analysis revealed no issues, we did not discuss it further in the discussion section.

[Lines 197–200] In the power analysis, the model type was set to a random-effects model with high heterogeneity, probability of Type II error = 20% (test power needed to ≥ 80%), an expected standardized mean difference (SMD) of 0.2, number of studies, and both sample sizes in the control and intervention groups.

[Lines 419–422] 3.5.1. Power analysis

The results of the power analysis showed that in all meta-analyses, meta-regressions, and subsequent sensitivity analyses, the test power was 100%, indicating that the sample size and number of studies were sufficient for each analysis.

Comment 9: Inadequate description of risk of bias assessment process

Response 9: We thank you for your kind comment. We apologize for not providing a detailed description of the risk of bias assessment process. As the process includes resolving disagreements, we have described this topic in Comment 10.

Comment 10: Limited explanation of reviewer disagreement resolution

Response 10: We thank you for your kind comment. Our approach was similar to the one described in a recent paper published in Behavioral Sciences (Peng et al., https://www.mdpi.com/2076-328X/14/12/1143), which states, “Any discrepancies encountered during the coding process were resolved through group discussions, with a third author acting as a judge when necessary.” In our research, the process of quality assessment and BCT coding were first conducted by one reviewer, and then the results were confirmed by another reviewer. “If discrepancies arise, both reviewers discussed and exchanged their views, and if the discussion could not resolve the issue, the third or fourth reviewer was consulted until a consensus was reached and the result was adopted.” The revisions are described as follows [Lines 140–144, 150–154]:

[Lines 140–144]

Old: Therefore, the final decision on the overall risk of bias was the result of consensus dis-cussions among at least two reviewers. As mentioned above, any disagreement was resolved by discussion, and if necessary, the third or fourth reviewer was consulted (Y.S. or Y.N.).

Revised: Each study was initially assessed by one reviewer (J.W., J.K., T.T., or R.M.), and the assessment results were confirmed by another reviewer (J.W., J.K., T.T., or R.M.). If there was a disagreement between the two reviewers, the result was finalized only after a discussion led to a consensus. If the disagreement could not be resolved, a third or fourth reviewer (Y.S. or Y.N.) was consulted.

[Lines 150–154]

Old: J.W., J.K., T.T., and R.M. conducted data extraction, which was confirmed by at least one reviewer (J.W., J.K., or Y.S.) who was not involved in the initial extraction. When two reviewers had different decisions on the extracted data, disagreements were resolved by discussing with a third reviewer or a fourth reviewer if it was necessary (Y.S. or Y.N.).

Revised: J.W., J.K., T.T., and R.M. conducted data extraction and BCT coding, which were confirmed by another reviewer (J.W. or J.K.) who was not involved in the initial extraction. When two reviewers had different decisions on the extracted data, disagreements were resolved by discussing with a third reviewer or a fourth reviewer (Y.S. or Y.N.) until a consensus was reached.

Comment 11: No mention of protocol pre-registration

Response 11: We thank you for your kind comment. In the manuscript, we have clarified that the study protocol was pre-registered in PROSPERO (pre-registered on April 30, 2022). In the Round 1 revision discussions, we addressed the issue regarding the delay in updating the registration information in the Methods and study limitations in the Discussion sections [Lines 102–105, 666–667]. We sincerely appreciate it if you could offer more specific guidance to help us confirm our response aligns with your comments.

[Lines 102–105] The protocol was initially registered in the International Prospective Register of Systematic Reviews (PROSPERO) on April 30, 2022, and revised on October 30, 2024, to include BCT extraction to explore effective components (CRD42022322275).

[Lines 665–666] the lack of timely updates in PROSPERO regarding revision to the research methods has reduced the credibility of this research;

Comment 12: Training for BCT coding not assessed

Response 12: We thank you for your kind comment. We have stated in the manuscript that the data extraction [Lines 149–150], including BCT coding, was not evaluated, and this limitation has also been addressed in the study limitations [Lines 659–660].

[Lines 149–150]  

Old: No assessment for training was conducted.

Revised: No assessment for training in data extraction, BCT coding or inter-rater reliability was conducted.

[Lines 658–659]

Old: the lack of assessment in training on data reduced the accuracy of the extractions.

Revised: the lack of assessment in training on data and BCT coding reduced the accuracy of the extractions. The lack of an inter-rater reliability assessment weakens the reproducibility of this study;

Comment 13: Very high heterogeneity (I² = 92%) not adequately addressed

Response 13: We thank you for your kind comment. As this study primarily aimed to explore which BCTs are related to effectiveness in promoting physical activity rather than exploring what influencing factors are, we only summarized the results in 4.4 Quality of the evidence and stated a conclusion that other influencing factors exist [Lines 625–633]. In the study limitations, we emphasized that unexplainable heterogeneity is a limitation of the research [Lines 672–673]. Nonetheless, understanding heterogeneity helps to comprehend other influencing factors, and we have outlined some potential factors and frameworks for analysis in future directions [Lines 704–725].

[Lines 625–633]

Old: The heterogeneity was attributed to differences in intervention components, which provided a basis for meta-regression analysis to explore the effectiveness of specific components.

Revised: The high heterogeneity among the included studies was identified in most analyses (Supplementary Table S7), which increased the CI width in the random-effects models. This heterogeneity may be attributed to factors, including study duration, basic theory, and sample sizes; combining different intervention components is an important source of heterogeneity [16, 21]. Nevertheless, in the subgroup and subsequent sensitivity analyses, heterogeneity was high. In the meta-regression and subsequent sensitivity analyses, models showed a low coefficient of determination, indicating weak explanatory power for heterogeneity (Supplementary Table S6). This suggested that the heterogeneity was not mainly due to the type of PA or BCTs.

[Lines 672–673]

Added: The high level of unexplained heterogeneity suggests that the discussion of intervention effects in this study is insufficient.

[Lines 704–725] (Please review the manuscript)

Improving the methodological limitations recognized in this study is crucial for enhancing the quality of future research. To explore the sources of heterogeneity, previous studies have highlighted the importance of follow-up duration [27, 28, 165, 176], as well as including other factors related to heterogeneity, such as the number of BCTs unique in the intervention group [16] and the total number of BCTs [28]. As it was reported, these findings show differences compared to those in previous studies [21]. Compared to similar studies conducted with different populations, our research provides generalizable results and reveals that the effects of these BCTs may vary depending on the target population. Understanding which BCTs are generally effective and which are more influenced by specific population characteristics is crucial. This can help identify the source of heterogeneity and design more effective intervention programs tailored to different groups. It is essential not only to identify highly generalizable BCTs but also to explore factors influencing their effectiveness. There is a need for numerous validation experiments on individual BCTs or factorial experiments designed based on a multiphase optimization strategy, as these can generate more direct and reliable foundational evidence [177, 180]. Michie et al. integrated the following five levels of intervention characteristics using the word “Behavior Change Ontology:” target behaviors, BCTs, theoretical mechanisms, modes of delivery, and contexts [161]. This research explored the effects of BCTs on the target behavior (PA) within a specific context (healthy young and middle-aged adults) but did not examine the theoretical mechanisms and modes of delivery. Therefore, incorporating these two levels in future studies will be critical for developing effective intervention programs to promote PA and better explain the heterogeneity.

Comment 14: Clear publication bias present but inadequately explored

Response 14: We thank you for your kind comment. We have addressed the publication bias issue in meta-analyses through the trim-and-fill method or by removing outliers in sensitivity analyses. However, a method to resolve publication bias in meta-regression analyses is lacking. This has been described in the discussion section [Lines 612–621]. Regarding why publication bias occurred, studies with less favorable results (especially smaller studies) may not be published. Extensive discussion of publication bias would divert from the main focus of this study; therefore, we have not elaborated on it in the main text.

[Lines 613–621] The meta-analyses and -regressions in this research faced challenges due to publication bias. While sensitivity analyses of meta-analyses employing the trim-and-fill method or excluding outliers adjusted the results, they also led to the loss of information from several included studies. By excluding studies with high risk (for reducing sedentary behavior, studies with low and high risk), the sensitivity analysis of four meta-regressions showed no publication bias. However, the sensitivity analyses of meta-regressions for walking and leisure-related PA metric outcomes did not eliminate publication bias. Egger’s test is a widely used method for detecting publication bias and is recommended by Cochrane [31]. Conversely, it has been noted to produce false positives when heterogeneity is high [175].

Comment 15: Some meta-regressions based on too few studies (k < 10)

Response 15: We thank you for your kind comment. Meta-regressions for exercise behavior (k = 4) and work-related PA metric outcomes (k = 7) were not conducted. For leisure-related PA metric outcomes (k = 11) and sensitivity analyses (k = 10), no associated BCTs were identified, likely due to the limited number of included studies. For the sensitivity analysis for exercise behavior, the power analysis indicated low statistical power (21.41%). However, due to the limited number of included studies, meta-regression was not performed (only results of power analyses on conducted meta-regressions have been shown in the main text). Discussions on them have been added [Lines 668–670].

[Line 668–670]

Added: the limited number of studies in sub-group analyses for leisure-related PA metric outcomes, or the absence of meta-regression for exercise behavior and work-related PA metric outcomes, may result in the inability to effectively assess the association of BCTs;

Comment 16: Wide confidence intervals in several analyses

Response 16: We thank you for your kind comment. We consider that the wide confidence intervals may result from the high level of heterogeneity. We have described this in Comment 13.

Comment 17: Insufficient reporting of effect sizes

Response 17: We thank you for your valuable comment. The average effect sizes (SMD) for each analysis have been reported in the main text of 3.3. Meta-analyses and meta-regressions and Table 1. We would appreciate it if you could specify the insufficient items so we can make the necessary revisions.

Comment 18: Missing subgroup analyses details

Response 18: We thank you for your kind comment. We are unsure if you are referring to the absence of certain subgroup results in Table 2. We apologize for presenting only partial results; thus, the reason for this has been added to the footnote as follows [Lines 285–288]:

[Lines 285–288] Regression analyses were not conducted for exercise behavior and work-related PA metric outcomes due to the limited number of included studies. No associated BCTs were confirmed in the analysis for leisure-related PA metric outcomes, and the results are not presented in Table 2.

Comment 19: Inadequate discussion of heterogeneity sources

Response 19: We thank you for your kind comment. We discussed high heterogeneity in Comment 13.

Comment 20: Some interpretations exceed what the data supports

Response 20: We thank you for your kind comment. We are confident that our interpretations are based on the study findings and well-supported by relevant evidence. We would sincerely appreciate it if you could highlight specifically which interpretations exceed the scope supported by the data.

Comment 21: Clinical significance of effects not adequately discussed

Response 21: We thank you for your kind comment. Based on the findings of the study, we have added a hypothetical inference as follows: healthy young and middle-aged adults may tend to favor intervention programs that provide emotional support, self-management, and respect autonomy in 4.6. Practical implications and recommendations of Discussion section [Lines 674–684]. We would sincerely appreciate it if you could provide better insights.

[Lines 674–684]

Old: These study findings have important implications. First, as there was limited discussion on which intervention components are effective in increasing the general health of young and middle-aged adults' PA, the findings of this study addressed the gap in understanding effective intervention components for promoting PA in this population. Second, the results of meta-analyses demonstrate the low effectiveness of current inter-ventions, suggesting the need for continued improvement of most intervention methods. Third, the quantitative meta-regressions of intervention components’ associations offered reliable references for designing intervention methods.

Revised: These study findings have important implications. First, as there was limited discussion on which intervention components are effective in increasing the general health of young and middle-aged adults' PA, the findings of this study addressed the gap in understanding effective intervention components for promoting PA in this population. Second, the results of meta-analyses demonstrate the low effectiveness of current interventions, suggesting the need for continued improvement of most intervention methods. Third, the quantitative meta-regressions of intervention components’ associations offered reliable references for designing intervention methods. Fourth, this study’s findings suggest that healthy young and middle-aged adults tend to favor intervention programs that provide emotional support, self-management, and respect for autonomy.

Comment 22: Future research directions too vague

Response 22: We thank you for your kind comment. Based on our analysis results, we believe that more research on factors influencing heterogeneity should be conducted in the future (Comment 13). Understanding the factors that affect intervention outcomes or the effectiveness of BCTs can help develop better intervention programs [Lines 704–725].

Comment 23: Implementation considerations limited

Response 23: We thank you for your kind comment. As this study serves as a preliminary exploration and is not designed to provide practical implications, along with its methodological limitations, we recommend conducting confirmatory experiments rather than directly applying the findings of this study. We have added the description as follows [Lines 698–700]:

[Lines 698–700]

Added: We encourage small-scale confirmatory experiments to validate the findings of this study. However, researchers should interpret the results cautiously and incorporate their professional expertise when designing large-scale formal intervention programs.

Comment 24: BCT mechanisms not fully explored

Response 24: We thank you for your kind comment. As stated in Comment 5, there is a lack of studies on the mechanisms, and the current evidence does not enable more extensive discussion.

Comment 25: Limited discussion of methodological limitations

Response 25: We thank you for your kind comment. We have revised the limitations section based on your comments [Lines 648–673].

Comment 26: No adjustment for multiple comparisons

Response 26: We thank you for your kind comment. We did not perform any multiple comparisons. We would appreciate it if you could provide more detailed information (e.g., line numbers or relevant keywords).

Comment 27: Publication bias adjustment methods inadequate

Response 27: We thank you for your kind comment. In Comment 14, we have described the adjustment methods used in this study, which successfully addressed publication bias in the meta-analyses. However, currently, there is a lack of methods to adjust for publication bias in meta-regression analyses, which we have identified as one of the study's limitations. We would sincerely appreciate it if you could provide suggestions for better approaches to help us address this issue.

Comment 28: Inconsistent citation format

Response 28: We thank you for your kind comment. We used the reference citation format provided by the MDPI for EndNote. Upon inspection, we noticed that the formats of certain terms did not align with those of published articles in Behavioral Sciences. In the revised manuscript, we have corrected these inconsistencies. However, we will make further corrections according to the editorial office’s final instructions.

Comment 29: Missing DOIs in references

Response 29: We thank you for your kind comment. Since recent articles in Behavioral Sciences do not include DOIs, we initially did not include them. We have now added DOIs in this revision, and we will make further corrections according to the editorial office’s final instructions.